# Nonlinear sigma model description of deconfined quantum criticality in arbitrary dimensions

Da-Chuan Lu

*Department of Physics, University of California at San Diego, La Jolla, CA 92093*

In this paper, we propose using the nonlinear sigma model (NLSM) with the Wess-Zumino-Witten (WZW) term as a general description of deconfined quantum critical points that separate two spontaneously symmetry-breaking (SSB) phases in arbitrary dimensions. In particular, we discuss the suitable choice of the target space of the NLSM, which is in general the homogeneous space $G/K$, where $G$ is the UV symmetry and $K$ is generated by $\mathfrak{k} = \mathfrak{h}_1 \cap \mathfrak{h}_2$, and $\mathfrak{h}_i$ is the Lie algebra of the unbroken symmetry in each SSB phase. With this specific target space, the symmetry defects in both SSB phases are on equal footing, and their intertwinement is captured by the WZW term. The DQCP transition is then tuned by proliferating the symmetry defects. By coupling the $G/K$ NLSM with the WZW term to the background gauge field, the 't Hooft anomaly of this theory can be determined. The bulk symmetry-protected topological (SPT) phase that cancels the anomaly is described by the relative Chern-Simons term. We construct and discuss a series of models with Grassmannian symmetry defects in 3+1d. We also provide the fermionic model that reproduces the $G/K$ NLSM with the WZW term.

## I. INTRODUCTION

The continuous symmetry-breaking transitions are well described within the Landau-Ginzburg-Wilson (LGW) paradigm, the local order parameter acquires a non-zero expectation value in the spontaneous symmetry breaking (SSB) phase, and vanishes continuously when approaching the transition point. However, within the LGW paradigm, two SSB phases cannot be joined by a continuous transition but only first order or level crossing. Quantum effects open the possibility to have a such continuous transition, and the first explicit example is the deconfined quantum critical point (DQCP) between the VBS phase and Néel phase in 2+1d quantum magnet [1–4].

The ordinary symmetry breaking transition can be alternatively understood by proliferating the symmetry defects in the SSB phase to arrive at the disordered phase. This point of view is particularly useful to understand the DQCP - the symmetry defect in the VBS phase is decorated with the quantum number of the spin $\mathsf{SU}(2)$ symmetry, proliferating which will restore the lattice rotation symmetry but break the spin symmetry, and arrive at the Néel phase [5]. Decorated symmetry defects are also studied in other beyond LGW quantum transitions [6–8].

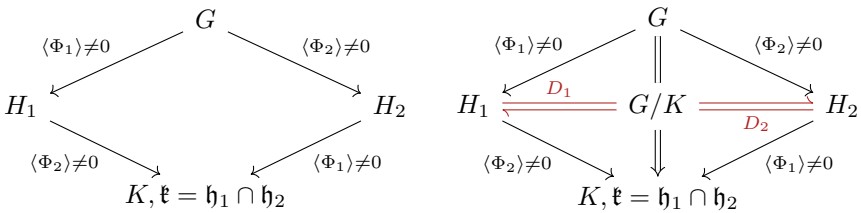

**FIG. 1:** $G$ is the UV symmetry and $K$ is generated by the Lie algebra $\mathfrak{k} = \mathfrak{h}_1 \cap \mathfrak{h}_2$, where $\mathfrak{h}_1, \mathfrak{h}_2$ are the Lie algebras of unbroken symmetries $H_1, H_2$ of SSB phases. The different SSB phases are obtained by condensing order parameters $\Phi_i$ as in the left graph. Alternatively, the right graph emphasizes using the symmetry defects in the homogeneous space $G/K$ where the bosonic field lives in. Proliferating symmetry defects $D_i$ will drive the transition from one SSB phase to the other due to the additional charges assigned by WZW term.

Intertwinement of symmetry defects in different SSB phases, i.e. the symmetry defect in one SSB phase carries the quantum number of the broken symmetry of the other SSB phase, is the prominent and ubiquitous feature of the DQCP theories. To properly describe the symmetry defects and their intertwinement, we consider the following generic symmetry breaking pattern of DQCP theory as shown in Fig. 1. Once condensing the order parameter $\Phi_i$, one arrives at different SSB phases with unbroken symmetry $H_i$. Further condensing the other order parameter, one arrives at the minimal symmetry $K$. We will show that the symmetry defects in $G/K$ incorporate all the symmetry defects in both SSB phases Sec. V. The reason is based on that the codimension-$(q+1)$ symmetry defect in each SSB phase is classified by $\pi_q(G/H_i)$ [9], and $\pi_\star(G/K)$ contains roughly the generators of these homotopy groups. Moreover, the generators of the homotopy group are related to the differential forms via the Hurewicz theorem, then these symmetry defects are described by the differential form in the Lagrangian, we may call this term as charge operator of symmetry defect. Technically, we use de Rham cohomology to find the generators of the cohomology group of $G/K$. The intertwinement of symmetry defects is essentially captured by the linking number of their corresponding charge operators, and the Wess-Zumino-Witten term in the action assigns phase to the linking number. Therefore, we propose using the nonlinear sigma model (NLSM) with the target space $G/K$ to describe the DQCP between two SSB phases with unbroken symmetry $H_1, H_2$, and the intertwinement of symmetry defects is described by the Wess-Zumino-Witten (WZW) term. We also use $G/K$ NLSM in short. The DQCP transition is driven by proliferating symmetry defects $D_i$ in $G/K$ along the red arrows in the right graph of Fig. 1. We will provide details and examples in the Sec. V.

Intertwinement of symmetry defects in different SSB phases is also the manifestation of the (mixed) 't Hooft anomaly of the global symmetry. When global symmetry has 't Hooft anomaly, the

theory is still well-defined unless the symmetry is gauged. The 't Hooft anomaly of global symmetry constrains the infrared phases not being trivial gapped phases but either SSB phase, gapless or topological order. The phase diagram of DQCP theory, namely two SSB phases connected by a gapless phase, agrees with the consequence of the 't Hooft anomaly. The anomaly of 1+1d, 2+1d DQCP theories have been carefully analyzed in [10–12]. However, previous anomaly analysis is based on gauge theories, we focus on the anomaly analysis of NLSM description in terms of coupling the theory to background gauge fields. Since $G$ is anomalous but $K$ is non-anomalous, the NLSM with target space $G/K$ saturates the anomaly of $G$. We provide a detailed calculation of coupling the WZW term to the background gauge field and show the gauged WZW term gives the 't Hooft anomaly. By anomaly inflow, the 't Hooft anomaly is canceled by one higher dimensional bulk (relative) Chern-Simons term [13–15] and discussed in Sec. IV. The lattice models that describe DQCP do not need additional higher dimensional bulk, it is because some of the global symmetries are acting in the *non-onsite* way, when it flows to IR, the field theory description requires these symmetries to be internal but with the 't Hooft anomaly [16, 17]. We should point out that the NLSM has long been used to describe the Goldstone mode in the SSB phase [18–20], and the additional Wess-Zumino-Witten term is used to match the anomaly in the ultraviolet [21]. Recently, the gauge theory and its dual NLSM with Stiefel manifold or Grassmannian manifold as the target space have been studied in the context of spin liquid and quantum critical point beyond the LGW paradigm [22–24]. In current paper, we present a general construction of NLSM with WZW that describes any given DQCP with continuous symmetry breaking. In the following, we explore its connection to the mixed 't Hooft anomaly, relative Chern-Simons term, and the intertwinement of different symmetry defects.

Inspired by recent work on deconfined quantum criticality (which can be critical point or critical phase depending on the model details) among grand unified theories [25, 26], we apply our framework to construct the theory of 3+1d deconfined quantum critical phase (DQCPh) with global symmetry $G = \mathsf{SO}(2n)$ that separates two SSB phases with unbroken symmetries $H_1 = \mathsf{U}(n), H_2 = \mathsf{SO}(2n-2m) \times \mathsf{SO}(2m)$, and $K = \mathsf{SU}(n-m) \times \mathsf{U}(1) \times \mathsf{SU}(m) \times \mathsf{U}(1)$. The symmetry defects are then described by $\pi_2(G/K) = \ker(\pi_1(K) \to \pi_1(G)) = \mathbb{Z} \oplus \mathbb{Z}$. This is particularly interesting since the symmetry defect in the SSB phase with unbroken symmetry $H_2$ is Grassmannian manifold and has topological charge $\pi_2(G/H_2) = \mathbb{Z}_2$, which cannot be captured by de Rham cohomology, but once embedding into the larger space $G/K$, the topological charge becomes integer, and it has corresponding differential form via de Rham cohomology. This manifests the similar way that the non-perturbative $\mathsf{SU}(2)$ anomaly (due to $\pi_4(\mathsf{SU}(2)) = \mathbb{Z}_2$) can be

perturbatively found by embedding $\mathsf{SU}(2) \hookrightarrow \mathsf{SU}(3)$ [21, 27], and here we embed $G/H_2 \hookrightarrow G/K$. This series of 3+1d DQCPh theories has "new $\mathsf{SU}(2)$ anomaly" of the global symmetry $\mathsf{SO}(2n)$ [28], and it is matched by symmetry-protected topological phase in 5d bulk described by $w_2 w_3(\mathsf{SO}(2n))$ [25, 26]. The mixed anomaly is obtained by pull-back via the embedding map of the subgroup into $G$.

We then present the alternative fermionic model that reproduces the $G/K$ NLSM with WZW term. The fermions are coupled to the fluctuating bosonic fields which live in the homogeneous space, the bosonic fields parameterize the mass manifold of the fermions. We dub such fermionic construction of the nonlinear sigma model as fermionic sigma model [29]. When integrating out the massive fermions, the effective action is the nonlinear sigma model with level-1 WZW term [29, 30]. For generic homogeneous space $G/K$, the fermion mass manifold needs to be properly chosen. This fermionic model also implies that the nonlinear sigma model with level-1 WZW term needs a spin structure which is used to define the parallel transport of spinor fields, though the Goldstone boson fields are bosonic [31]. We then construct the fermionic sigma model of the 4d DQCPh theories and explicitly show the charge operators of two symmetry defects in different SSB phases link together.

The paper is structured as follows, we review the essential ingredients of the nonlinear sigma model and Wess-Zumino-Witten term as well as Lie group cohomology in Sec. II. Then we review the 't Hooft anomaly and anomaly matching by WZW term in Sec. III. Readers who are familiar with these can safely skip Sec. II and Sec. III but skimming through the notations would be helpful. We present the gauged WZW term and its anomaly matching with the bulk (relative) Chern-Simons term for generic spontaneously symmetry breaking in Sec. IV. We construct specific DQCP theories in Sec. V and present the fermionic sigma model description in Sec. VI. We summarize our results and list further directions in Sec. VII. Finally, there are two appendices about de Rham cohomology of Lie group in Appendix. A and Cartan homotopy method in Appendix. B that are used to derive an explicit formula for the various WZW terms and Chern-Simons terms.

## II. REVIEW OF NONLINEAR SIGMA MODEL AND WESS-ZUMINO-WITTEN TERM FOR GENERAL HOMOGENEOUS SPACE $G/H$

### A. The Lagrangian of NLSM and WZW term

Supposing the UV theory has global symmetry $G$, which can contain spacetime symmetry as well as internal symmetry. In the IR, the symmetry is spontaneously broken down to $H$, then the

IR theory contains the part with unbroken global symmetry $H$ and the gapless Goldstone mode lives on the coset $G/H$. For example, the 3+1d $N_f$ flavors fermion couples to $\mathsf{SU}(n)$ gauge field, the global symmetry $\mathsf{SU}(N)_L \times \mathsf{SU}(N_f)_R$ is spontaneously broken down to $\mathsf{SU}(N)_{\text{diag}}$, the coset where the Goldstone boson lives in is simply the Lie group $\mathsf{SU}(N)$ [21]. Or the Heisenberg spin in 2+1d, the spin rotation symmetry $\mathsf{SO}(3)_S$ is spontaneously broken down to $\mathsf{SO}(2)_S$, and the Goldstone mode lives in the coset $\mathbb{S}^2_S = \frac{\mathsf{SO}(3)_S}{\mathsf{SO}(2)_S}$ [32]. Before parametrizing the coset $G/H$, let's take the Goldstone boson lives in an arbitrary closed manifold $M$.

The Goldstone bosons are described by the nonlinear sigma model, where the scalar field takes value in the target manifold $M$. The field configuration is represented by a map from $d$-dimensional spacetime manifold $X$ to the target manifold $M$,

$$U(x^\mu) : X \to M, \tag{II.1}$$

where $x^\mu$ is the coordinate of $X$ and $U$ lives in $M$. The kinetic term is,

$$S_0 = \frac{1}{2} \int_X d^d x \, \text{tr}(U^{-1}\partial_\mu U U^{-1} \partial^\mu U). \tag{II.2}$$

where repeated indices mean summation. Besides the kinetic term, one can define the WZW term by pull-back the closed form $\Gamma^{(d+1)}$ on $M$. It seems the WZW term depends on the additional dimension, however, since the variation of the closed $(d+1)$-form yields the exact form, $\delta\Gamma^{(d+1)} = d\eta^{(d)}$, according to the Stokes' theorem, the equation of motion is indeed in $d$-dimension and does not rely on the fictitious extra dimension. We may view the spacetime manifold $X$ as the boundary of a certain bulk manifold $Y$, such that $\partial Y = X$, and extend the map $\tilde{U} : Y \to M$, the WZW action is,

$$S_{\text{WZW}} = 2\pi k\mathbf{i} \int_Y \tilde{U}^*(\Gamma^{(d+1)}), \tag{II.3}$$

where $\tilde{U}^*$ is the pull-back map, $k$ is the quantized level which is important for the theory to be well-defined and not depend on an extension to the bulk $Y$. Suppose we have another extension $\bar{Y}$ which is the orientation reverse of $Y$, and $Y \cup \bar{Y}$ is a closed manifold, then the integral over this combined manifold should be $2k\pi\mathbf{i}$, $k \in \mathbb{Z}$, such that the WZW term does not depend on the extension, otherwise, there is a phase ambiguity for different extensions. $\Gamma^{(d+1)}$ is actually the generator of integral cohomology of $M$, $\Gamma^{(d+1)} \in H^{(d+1)}(M, \mathbb{Z})$. If the closed form is also exact, then the WZW term is a term expressed in the $d$-dimensional spacetime.

Apart from the closed $(d+1)$-form which can be used to define the WZW term, other closed $q$-form with $q < d$ can be used to represent the charge operators of the possible topological defects.

The topological defects are classified by the homotopy group [9], and the $q$th homotopy group of $M$ is isomorphic to the homology group of $M$ via the Hurewicz theorem if $M$ is $(q-1)$-connected. Therefore, the charge operator of possible codimension-$(q+1)$ topological defect is given by the generator of $H^q(M, \mathbb{Z})$. This charge operator is like a counter, if the defect matches, then yield 1, and 0 otherwise. For example, the baryon in 4d $\mathsf{SU}(N)$ gauge theory is classified by $\pi_3(SU(N_f)) = \mathbb{Z}$, the charge operator or the baryon number current is $\Gamma^{(3)} \in H^3(\mathsf{SU}(N_f), \mathbb{Z})$, and it couples to the $\mathsf{U}(1)$ background gauge field $A$ as,

$$\exp\left(\mathbf{i} \int_{X^4} A \wedge \Gamma^{(3)}\right) \tag{II.4}$$

This reproduces the Goldstone-Wilczek current [33, 34].

To sum up, the WZW term and the charge operators of the topological defects in SSB phases are given by the generators of the integer coefficient cohomology group of the target space with a certain degree. In the following, we are using de Rham cohomology to find these generators. Since the coefficient of de Rham cohomology is $\mathbb{R}$, one needs to normalize these generators such that the integral on the generator of the corresponding homotopy group yields 1 [31]. After normalization, this gives the generators of the cohomology group with $\mathbb{Z}$ coefficient and can be used to define the WZW term and charge operators of the topological defects.

### B. Construction of the coset

The Goldstone boson field $U$ lives in the coset $G/H$, meaning that the Goldstone boson fields are equivalent under the right multiplication of the elements in $H$, $U \sim U'h$, $h \in H$. We need the following parametrization of the coset $G/H$ [18]. We denote the generators of compact Lie group $G$ as $T^A, A = 1, ..., \dim G$, and the subgroup $H$ has generators $T^\alpha, \alpha = \dim G - \dim H + 1, ..., \dim G$. The orthogonal part of the $\mathfrak{h}$ in $\mathfrak{g}$ is $\mathfrak{f} = \mathfrak{g} - \mathfrak{h}$, denoted as $T^a, a = 1, ..., \dim G - \dim H$ (capital letters are for generators in $\mathfrak{g}$, greek letters are for those in $\mathfrak{h}$ and lower case letters are for $\mathfrak{f}$). We have grouped the indices such that $\mathfrak{g} = \mathfrak{f} \oplus \mathfrak{h}$. These generators in general satisfy the algebraic relation $[\mathfrak{h}, \mathfrak{h}] \subset \mathfrak{h}$, $[\mathfrak{h}, \mathfrak{f}] \subset \mathfrak{f}$, $[\mathfrak{f}, \mathfrak{f}] \subset \mathfrak{g}$. We often encounter that the coset $G/H$ is a symmetric space, in this case, the relation is, $[\mathfrak{h}, \mathfrak{h}] \subset \mathfrak{h}$, $[\mathfrak{h}, \mathfrak{f}] \subset \mathfrak{f}$, $[\mathfrak{f}, \mathfrak{f}] \subset \mathfrak{h}$. For $\mathfrak{h} = \varnothing$, the coset is simply $G$. The Goldstone boson field $U(\pi(x))$ is parametrized by the Nambu-Goldstone boson $\pi^a(x)$ as [19, 20, 35],

$$U(\pi) = e^{\mathbf{i}\pi^a(x)T^a}, \quad T^a \in \mathfrak{f}. \tag{II.5}$$

A general element $g$ in $G$ acting on the coset $U(\pi)$ gives,

$$g^{-1}U(\pi) = e^{i\pi'^a(\pi,g)T^a}e^{i\lambda_\alpha(\pi,g)T^\alpha} = U(\pi')h^{-1}(\pi,g) \tag{II.6}$$

the group transformation is equivalent to $g : U(\pi) \to U(\pi') = g^{-1}U(\pi)h(\pi,g)$, where $h(\pi,g)$ as well as $\pi$ depend on the spacetime coordinate. $\pi(x)$ is in general transformed in a complicated nonlinear way. But when restricted to $H$, one can always choose the Goldstone boson $\pi(x)$ transformed in a linear way.

### C. Cohomology of the homogeneous space

The generators of the cohomology group are particularly relevant to the terms that describe the symmetry defects and the WZW term. The cohomology of homogeneous space $G/H$ is given by the closed $G$-invariant forms on $G/H$ modulo exact $G$-invariant forms. We first discuss differential forms on $G$ and then restrict them on $G/H$. These differential forms are constructed by the basis of left-invariant 1-forms, the Maurer–Cartan 1-form on $G$ [1],

$$\theta \equiv U^{-1}(\pi)\mathrm{d}U(\pi) = \theta^A T^A. \tag{II.7}$$

where $T^A$ is the Lie algebra generator and $\theta^A$ is the component. The Maurer-Cartan 1-form is Lie-algebra valued 1-form on $G$, and its component satisfies the Maurer-Cartan equations,

$$\mathrm{d}\theta^C = -\frac{1}{2}f^{ABC}\theta^A \wedge \theta^B, \tag{II.8}$$

where $f^{ABC}$ is the structure constant of the Lie group. Then the general left-invariant $n$-form on $G$ is given by, $\Omega_G^{(n)} = \frac{1}{n!}(\Omega_G)_{A_1,...,A_n}\theta^{A_1} \wedge ... \wedge \theta^{A_n}$. If the left-invariant $n$-form on $G$ is closed, $d\Omega_G^{(n)} = 0$, but non-exact, $\Omega_G^{(n)} \neq d\eta_G^{(n-1)}$, then it gives the generator of the cohomology groups of $G$.

On the other hand, the invariant $n$-form on $G/H$ should satisfy, a) the indices vanish on $\mathfrak{h}$, and b) invariant under the adjoint action of $\mathfrak{h}$. These two conditions can be explicitly expressed using the component of Maurer-Cartan 1-form, namely,

$$\Omega^{(n)} = \frac{1}{n!}\Omega_{a_1,...,a_n}\theta^{a_1} \wedge ... \wedge \theta^{a_n}, \tag{II.9}$$

$$\mathcal{L}_\alpha \Omega^{(n)} = -\sum_{i=1}^{n}\frac{1}{n!}\Omega_{a_1,...,a_n}f^{b_j,\alpha,a_j}\theta^{a_1} \wedge ... \wedge \theta^{b_j} \wedge ... \wedge \theta^{a_n} = 0. \tag{II.10}$$

---

[1] The Maurer–Cartan 1-form is on $G$ instead of $G/H$, additional conditions need specifying as discussed later. More details can be found in, for example, [36]

Then the cohomology of $G/H$ is given by,

$$H^*(G/H, \mathbb{R}) = \frac{\text{invariant closed } n\text{-form on } G/H}{\text{invariant exact } n\text{-form on } G/H}. \tag{II.11}$$

Note that the de Rham cohomology of $G/H$ is isomorphic to the relative Lie algebra cohomology, $H^*(G/H, \mathbb{R}) = H^*(\mathfrak{g}, \mathfrak{h}; \mathbb{R})$, therefore, we use de Rham cohomology of $G/H$ and relative Lie algebra cohomology interchangeably.

It is convenient to decompose the $\mathfrak{g}$-valued 1-form $\theta$ into $\mathfrak{h}$-valued and $\mathfrak{f}$-valued parts,

$$\theta = U^{-1}dU = (U^{-1}dU)|_{\mathfrak{f}} + (U^{-1}dU)|_{\mathfrak{h}} \equiv \phi + V, \tag{II.12}$$

and in the component form, $\theta = \theta^A T^A = \theta^a T^a + \theta^\alpha T^\alpha = \phi + V$. The above conditions can be intuitively understood by doing the group action on the 1-forms according to Eq. (II.6),

$$\theta \to h^{-1}\theta h + h^{-1}dh, \tag{II.13}$$

$$V \to h^{-1}Vh + h^{-1}dh, \quad \phi \to h^{-1}\phi h, \tag{II.14}$$

therefore, $\phi = \theta^a T^a$ transforms under the adjoint action of $\mathfrak{h}$, while $V = \theta^\alpha T^\alpha$ transforms as the $\mathfrak{h}$-valued connection. The invariant $n$-form on $G/H$ is then given by the combination of $\phi$ and curvature $W = dV + V \wedge V$ or using the component form under the condition Eq. (II.9).

### D.   Generators of cohomology group on $G/H$

Among these invariant $n$-forms on $G/H$, the cohomology on $G/H$ is obtained by closed invariant $n$-form modulo invariant exact $n$-form. We postpone the detailed algorithm that finds the generators of the cohomology group to Appendix. A. For physical relevance, we are interested in the generator with a degree less than 6, for example, the degree 5 generator may correspond to the 4d WZW term, and the degree 3 generator corresponds to 2d WZW term.

*a.   Compact Lie group $G$*   For $H = \varnothing$, the cohomology group of $G$ has degree 3 generator for all compact Lie group ($\mathsf{SU}, \mathsf{SO}, \mathsf{Sp}$), and degree 5 generator only for $\mathsf{SU}$ group [37]. The generators are given by,

$$x^{(3)} = \frac{1}{3}\text{tr}(U^{-1}dU)^3 = \frac{1}{3}\text{tr}\theta^3, \quad x^{(5)} = \frac{1}{10}\text{tr}(U^{-1}dU)^5 = \frac{1}{10}\text{tr}\theta^5. \tag{II.15}$$

These generators are of cohomology group with $\mathbb{R}$ coefficient, and they need to be normalized such that the integral over the generator of $\pi_3(G) = \mathbb{Z}$, $\pi_5(\mathsf{SU}) = \mathbb{Z}$ equal to 1 [31, 38]. The

normalized forms are the generators of $H^3(G,\mathbb{Z}), H^5(\mathsf{SU},\mathbb{Z})$. These generators can be written in the component form as,

$$x^{(3)} = \frac{1}{6}f_{ABC}\theta^A \wedge \theta^B \wedge \theta^C, \tag{II.16}$$

$$x^{(5)} = \frac{1}{40}d_{A_1BC}f_{BA_2A_3}f_{CA_4A_5}\theta^{A_1} \wedge \theta^{A_2} \wedge \theta^{A_3} \wedge \theta^{A_4} \wedge \theta^{A_5} \tag{II.17}$$

where $f_{ABC} = \mathrm{tr}(T^A[T^B, T^C])$ is the structure constant, $d_{ABC} = \mathrm{tr}(T^A\{T^B, T^C\})$ is the totally symmetric rank 3 tensor. The totally symmetric rank 3 tensors are non-zero for $\mathsf{SU}(N)$ group with $N > 2$.

   b.   *Homogeneous space $G/H$*   The cohomology of homogeneous space is much richer, the generators in general should satisfy Eq. (II.9). Before case-by-case discussion of the homogeneous spaces, the non-trivial generators of $H^3(G/H,\mathbb{Z}), H^5(G/H,\mathbb{Z})$ that may correspond to 2d and 4d WZW terms or codimension-4, 6 symmetry defects are given by,

$$y^{(3)} = \frac{1}{3}\mathrm{tr}(3\phi W + \phi^3), \quad y^{(5)} = \frac{1}{5}\mathrm{tr}(\phi^5 + \frac{10}{3}W\phi^3 + 5\phi W^2), \tag{II.18}$$

where $W = dV + V \wedge V$ is the curvature of $V$. Since $V$ transforms as $\mathfrak{h}$-valued connection based on Eq. (II.13), its curvature will transform as adjoint action under $H$ as well as $\phi$. The generators $y^{(3)}, y^{(5)}$ are invariant under $G$. One can express these generators in terms of Goldstone boson field by $\phi = (U^{-1}dU)|_{\mathfrak{f}}, V = (U^{-1}dU)|_{\mathfrak{h}}$. We postpone the derivation of these generators to the discussion of its corresponding anomaly.

   Similarly, we can express these generators in terms of components,

$$y^{(3)} = \frac{1}{6}(d_{ad}f_{dbc}\theta^a \wedge \theta^b \wedge \theta^c - 2d_{a\beta}f_{\beta bc})\theta^a \wedge \theta^b \wedge \theta^c \tag{II.19}$$

$$y^{(5)} = \frac{1}{60}(3d_{a_1bc}f_{ba_2a_3}f_{ca_4a_5} - 4d_{a_1b\gamma}f_{ba_2a_3}f_{\gamma a_4a_5}$$
$$+ 8d_{a_1\beta\gamma}f_{\beta a_2a_3}f_{\gamma a_4a_5})\theta^{a_1} \wedge \theta^{a_2} \wedge \theta^{a_3} \wedge \theta^{a_4} \wedge \theta^{a_5} \tag{II.20}$$

where the lower case letters are for $\mathfrak{f}$ part, and the Greek letters are for $\mathfrak{h}$ part. $d_{ab} = \mathrm{tr}(\{T^a, T^b\})$ is the totally symmetric rank-2 tensor, which is proportional to Kronecker delta for the most cases.

   The 5th cohomology groups are non-vanishing for $\mathsf{SU}(n)/\mathsf{SO}(n), n \geq 3$ and $\mathsf{SU}(2n)/\mathsf{Sp}(n), n \geq 2$ which are relevant to the spontaneous symmetry breaking of QCD with $\mathsf{SO}$ gauge group and $\mathsf{Sp}$ gauge group. More details can be found in Appendix. A 3.

   Besides the generators that correspond to WZW terms, there are low degree cohomology generators corresponding to the charge operators of topological defects. The second cohomology group is of particular interest in the following specific models since the generators of it correspond to the

charge operators of codimension 3 topological defects, they are particle-like in 3d and string-like in 4d. The second cohomology on $G/H$ is related to the first Chern class, evaluated on $\mathfrak{h}$-valued gauge field on $G/H$ [36]. In terms of the 1-forms, the generator of $y^{(2)} = H^2(G/H, \mathbb{R})$ is given by,

$$y^{(2)} = m_{\alpha_0} W^{\alpha_0} = \frac{1}{2} m_{\alpha_0} f^{\alpha_0 bc} \theta^b \wedge \theta^c \tag{II.21}$$

where $\alpha_0$ is the index for the $\mathsf{U}(1)$ factor in $\mathfrak{h}$, if $\mathfrak{h}$ has the decomposition $\mathfrak{h} = \mathfrak{h}_1 + ... + \mathfrak{u}(1) + ....$ The elements in $\mathfrak{h}_i$ vanish similar to that the first Chern class of non-abelian gauge field vanishes. We will demonstrate this explicitly in the following specific models.

The 4th cohomology is constructed in a similar way by using the symmetric tensor $m_{\alpha,\beta}$ that is invariant under the adjoint transformation of $H$,

$$y^{(4)} = m_{\alpha,\beta} W^\alpha \wedge W^\beta \tag{II.22}$$

The non-vanishing 2nd and 4th cohomology of some homogeneous spaces $G/H$ are listed in the Appendix. A 3 [37].

## III. REVIEW OF 'T HOOFT ANOMALY AND ANOMALY MATCHING BY WZW TERM

### A. 't Hooft anomaly and anomaly inflow

The theory with global symmetry that has 't Hooft anomaly is still well-defined but the anomalous symmetry cannot be gauged, otherwise, the anomaly is lifted to gauge anomaly and the theory is inconsistent. Recent understanding of symmetry-protected topological (SPT) phases gives a general picture of anomaly matching, the anomalous theory in $d$-dimension can be viewed as the boundary of $d+1$-dimension SPT (or invertible phase), and the anomaly is canceled by the bulk, therefore, the bulk-boundary combined system is non-anomalous [14, 39, 40].

The 't Hooft anomaly for global symmetry $G$ of a quantum field theory constrains the infrared phases to be

- gapless with $G$ symmetry
- spontaneously symmetry breaking
- topological order

but never a trivial gapped phase. The theory with 't Hooft anomaly is dubbed as "anomalous theory $\mathcal{T}$". The 't Hooft anomaly of the global symmetry in a theory can be found by coupling the theory to the background gauge field associated with its global symmetry. When performing

a gauge transformation on the background gauge field, the partition function of the anomalous theory on spacetime manifold $X$ instead of being invariant becomes,

$$\mathcal{Z}_{\mathcal{T}}[A + \delta_\lambda A] \to \mathcal{Z}_{\mathcal{T}}[A]e^{\mathbf{i} \int_X \alpha(\lambda, A)}. \tag{III.1}$$

where $\lambda$ is some gauge parameter. The partition function suffered from the ambiguity that different regularization yields different results. Some ambiguity can be cured by adding local counterterms, but for anomalous theory, the phase remains.

However, one can eliminate the ambiguity by viewing the anomalous theory $\mathcal{T}$ as the boundary of certain SPT phase $\mathcal{I}$. We can extend the background gauge field to the bulk $Y$, $\partial Y = X$, the partition function of the SPT phase under the gauge transformation is,

$$\mathcal{Z}_{\mathcal{I}}[A] = e^{-\mathbf{i} \int_Y \omega(A)} \to \mathcal{Z}_{\mathcal{I}}[A + \delta_\lambda A] = e^{-\mathbf{i} \int_Y \omega(A) - \mathbf{i} \int_X \alpha(\lambda, A)}. \tag{III.2}$$

Therefore, the bulk-boundary combined system is invariant under the gauge transformation,

$$\mathcal{Z}_{\mathcal{T}}[A]\mathcal{Z}_{\mathcal{I}}[A] \to \mathcal{Z}_{\mathcal{T}}[A + \delta_\lambda A]\mathcal{Z}_{\mathcal{I}}[A + \delta_\lambda A] = \mathcal{Z}_{\mathcal{T}}[A]\mathcal{Z}_{\mathcal{I}}[A] \tag{III.3}$$

The pictorial description is shown in Fig. 2. Using the bulk SPT to cancel the 't Hooft anomaly of the boundary theory is called anomaly inflow. On the other hand, the bulk SPT determines the 't Hooft anomaly of the boundary theory.

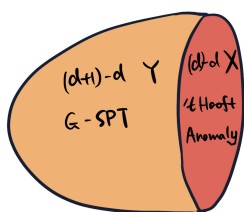

**FIG. 2:** Bulk-boundary combined system is invariant under the gauge transformation.

## B. Anomaly matching by Wess-Zumino-Witten term

The 't Hooft anomaly is a property of the Hilbert space, therefore, it should be matched in the ultraviolet and the infrared theory. From UV to IR, a common scenario is that the theory has spontaneously symmetry breaking. Suppose the UV theory $\mathcal{T}_{UV}$ has the global symmetry $G$, and it is spontaneously broken down to $H$ in the IR, the IR theory contains the $\mathcal{T}_{IR}$ with the unbroken $H$ symmetry and Goldstone bosons $U$ lives in the coset $G/H$. If the UV theory $\mathcal{T}_{UV}$ has an anomaly, then the IR theory $\mathcal{T}_{IR}$ together with the Goldstone boson $U$ should match the anomaly in $\mathcal{T}_{UV}$.

We can break the sufficient symmetry such that $\mathcal{T}_{IR}$ does not suffer from the anomaly and all the UV anomaly is matched by the Goldstone boson that lives in the homogeneous space $G/H$.

The chiral anomaly is an example that the local (perturbative) anomaly which can be seen from the triangle diagram is known to be matched by the Goldstone boson with WZW term [41, 42]. However, the global (non-perturbative) anomaly is more subtle and needs a proper definition for the WZW term [31, 43, 44]. In some cases, the non-perturbative anomaly can be found perturbatively by embedding the group into a larger group [21].

Coupling the WZW term to gauge field and constructing gauge invariant gauged WZW term has been extensively studied over the three decades [45–48], and it has very rich mathematical structures [49–51]. The gauged WZW term for general coset was studied in Ref. [52, 53].

In the following, we discuss a bulk-boundary combined construction of the gauged WZW term that could match general 't Hooft anomaly [15]. More specifically, assuming the $d$-dimensional UV theory has an anomaly described by $d+1$-dimension SPT phase $\mathcal{I}$, and the anomalous symmetry $G$ is spontaneously broken down to (anomalous or not) $H$ in the IR. Ref. [15] defines the general WZW term associated with $\mathcal{I}$ so that the anomalies of UV and IR are matched.

As presented in Sec. II B, the Goldstone boson lives in the coset $G/H$, where $G$ can contain both spacetime and internal symmetries. Another view of the coset $G/H$ is that the Goldstone boson locally takes value in $G$ and has gauge symmetry $H$, or cover of $H$. The Goldstone boson transforms under $G$ as Eq. (II.6). Consider the connection $A$ on the principal $G$-bundle, $A$ is the background gauge field associated with the global symmetry $G$, the gauge transformation of $A$ is given by,

$$A \to A^g = g^{-1}Ag + g^{-1}dg. \tag{III.4}$$

We can use the Goldstone boson field as the transition function to define the new connection,

$$A^U \equiv U^{-1}AU + U^{-1}dU \tag{III.5}$$

which transforms as $H$ connection under $G$ action according to Eq. (II.6),

$$A^U \to h^{-1}A^U h + h^{-1}dh. \tag{III.6}$$

Therefore, $A^U$ is the connection of the principal $H$-bundle. Note that the Lie algebra valued 1-form $\theta = U^{-1}dU$ is precisely $A^U|_{A=0}$. Similarly, we can decompose the connection into $\mathfrak{h}$ and $\mathfrak{f}$ part, they transform under $G$ action as,

$$A^U = A^U_{\mathfrak{h}} + A^U_{\mathfrak{f}} \to (h^{-1}A^U_{\mathfrak{h}}h + h^{-1}dh) + (h^{-1}A^U_{\mathfrak{f}}h) \tag{III.7}$$

where $A_{\mathfrak{h}}^U$ is the connection of $H$-bundle and $A_{\mathfrak{f}}^U$ transforms under the adjoint action of $H$. Since $A^U$ and $A_{\mathfrak{h}}^U$ are the connections of the same bundle, we can consider the interpolation of these connections,

$$A^U(t) = A_{\mathfrak{h}}^U + (1-t)A_{\mathfrak{f}}^U = \begin{cases} A^U & t = 0 \\ A_{\mathfrak{h}}^U & t = 1 \end{cases} \tag{III.8}$$

As shown in Fig. 3, one can imagine a cylinder where the leftmost is the connection $A^U$, and the rightmost is the connection $A_{\mathfrak{h}}^U$. The leftmost gauge field $A^U$ is extended to the bulk SPT with anomalous symmetry $G$ via the transition function $U$, while the rightmost gauge field $A_{\mathfrak{h}}^U$ is extended to the bulk with anomalous symmetry $H$ (if $H$ is non-anomalous, then the extension is not needed).

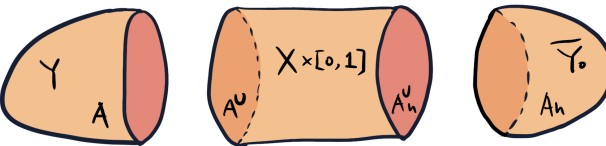

**FIG. 3:** Pictorial description of the construction of WZW term. The left manifold $Y$ describes the SPT with symmetry $G$ whose connection is the background gauge field $A$. With the transition function $U$, the left manifold $Y$ can be glued with the middle cylinder $[0,1] \times X$, where the WZW term lives in. The path $[0,1]$ connects the connection $A^U$ and $A_{\mathfrak{h}}^U$, where the background gauge field $A_{\mathfrak{h}}^U$ can be extended to the right manifold $\bar{Y}$ with global symmetry $H$. The WZW term on the cylinder then describes the Goldstone boson in the symmetry breaking phase with $G \to H$.

The general gauged WZW term in [15] is defined by taking the partition function of the invertible theory on the total manifold $Y_{\text{total}} = Y \cup (X \times [0,1]) \cup \bar{Y}_0$. The resulting partition function is gauge invariant. The Goldstone boson field $U$ is defined on the cylinder $X \times [0,1]$, therefore, the WZW term actually only depends on the dynamics of $U$ on $d$-dimensional manifold $X$. The connection $A^U$ at the left-most of the cylinder is extended to $Y$ by transition function $U$, $A_{\mathfrak{h}}^U$ is extended to $\bar{Y}_0$.

However, this construction is slightly different from the ordinary understanding of the WZW term, namely the WZW term only depends on the spacetime manifold $X$, though it is written in one higher dimension. In the following section, we provide an alternative construction of the gauged WZW term that is in accordance with the pictorial understanding of anomaly inflow in Fig. 2

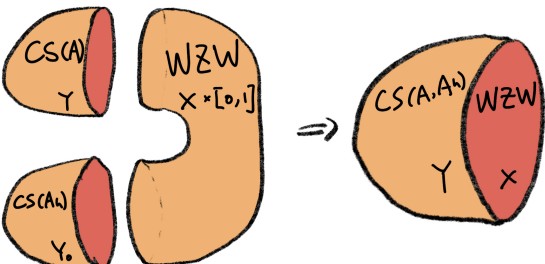

**FIG. 4:** Instead of defining WZW term on a cylinder as discussed in Fig. 3, we bend the cylinder such that the anomaly matching agrees with the gauge invariant bulk-boundary combined system in Fig. 2, and the WZW term only depends on spacetime manifold $X = \partial Y$. However, the bulk SPT is now described by relative Chern-Simons term.

## IV. (RELATIVE) CHERN-SIMONS FUNCTIONAL, GAUGED WZW TERM AND ITS ANOMALY

We first recall the setup of our system - given an anomalous UV global symmetry $G$, and it is spontaneously broken down to anomalous or not symmetry $H$ in the IR, then the UV anomaly is matched by the Goldstone boson in $G/H$ and IR theory with unbroken symmetry $H$. Instead of the construction in Sec. III B, we give an alternative construction of gauged WZW term that agrees with the anomaly inflow picture in Fig. 2, however, as shown in Fig. 4, the price is that the bulk SPT is described by the more complicated relative Chern-Simons functional (for $H = \varnothing$, the relative Chern-Simons term reduces to the Chern-Simons term).

For two connections $A$, $A'$ on the principle $H$-bundle, and a path of connection that interpolates these two, the *relative* Chern-Simons term is defined by the form that trivializes the difference between the curvature characteristic forms of different connections [13],

$$d\mathsf{CS}^{(2n+1)}(A, A') = \mathsf{ch}^{(2n+2)}(A) - \mathsf{ch}^{(2n+2)}(A') \tag{IV.1}$$

where $\mathsf{ch}^{(2n)}(A) = \frac{1}{n!}\mathrm{tr}(\mathbf{i}F/(2\pi))^n$, $F = dA + A \wedge A$ is the curvature of the connection $A$. The Chern-Simons form is then the special case of the relative Chern-Simons form with $A' = 0$, $d\mathsf{CS}^{(2n+1)}(A) = \mathsf{ch}^{(2n+2)}(A)$.

As reviewed in the previous section, the theory with the 't Hooft anomaly needs to be matched with bulk SPT. The SPT $\mathcal{I}$ with $\mathsf{U}$ or $\mathsf{SU}$ symmetry can be expressed as Chern-Simons functional, and many other SPTs can be obtained by higgsing the gauge field, for example, discrete gauge theories can be obtained from the Chern-Simons functional [54, 55]. We take the SPT $\mathcal{I}$ to be

described by the Chern-Simons functional or more general the relative Chern-Simons functional [15, 56],

$$\mathcal{Z}_{\mathcal{I}}[A] = \exp\left(\mathbf{i}k \int_Y \mathsf{CS}(A)\right), \quad \mathcal{Z}_{\mathcal{I}}[A, A'] = \exp\left(\mathbf{i}k \int_Y \mathsf{CS}(A, A')\right). \tag{IV.2}$$

where $k \in \mathbb{Z}$ is the level, $k = 1$ for SPTs. The anomaly associated with these SPTs is matched by the gauged WZW term in the following form,

$$\underline{\Gamma}^{(d+1)}(U, A, A_{\mathfrak{h}}) \equiv \mathsf{CS}(A^U, A_{\mathfrak{h}}^U) - \mathsf{CS}(A, A_{\mathfrak{h}}) = \Gamma^{(d+1)}(U) + d\alpha^{(d)}(U, A, A_{\mathfrak{h}}), \tag{IV.3}$$

where $\alpha^{(d)}(U, A, A_{\mathfrak{h}})$ is a $d$-form, and it is clear that the gauged WZW term does not depend on the extra dimension, since the first term is a closed $(d+1)$ form whose variation depends on the $d$-dimensional boundary $X$ and the second term only depends on the boundary $X$ by the Stokes' theorem. The relative Chern-Simons form $\mathsf{CS}(A, A')$ manifests the gauge invariance under $H$ transformation, and the gauged WZW term is invariant under $U \to Uh$ without any counterterms. If $\mathfrak{h} = \varnothing$, then the gauged WZW term is given by the Chern-Simons form,

$$\underline{\Gamma}^{(d+1)}(U, A) \equiv \mathsf{CS}(A^U) - \mathsf{CS}(A) = \Gamma^{(d+1)}(U) + d\alpha^{(d)}(U, A). \tag{IV.4}$$

We note that the gauged WZW term in Eq. (IV.3) indeed reproduces the 't Hooft anomaly under the global symmetry $G$. Let's first focus on the case when $\mathfrak{h} = \varnothing$, the connection $A \to g^{-1}Ag + g^{-1}dg$ and Goldstone boson field $U \to g^{-1}U$ under the $G$ transformation. $A^U$ is then invariant under this transformation, therefore, the first term $\mathsf{CS}(A^U)$ is invariant under the gauge transformation of $G$. However, the second term $-\mathsf{CS}(A)$ contributes to the anomalous phase under the transformation of $G$ by the descent equation argument.

For the case when $\mathfrak{h} \neq \varnothing$, under the $G$ transformation, the connection transforms as $A \to g^{-1}Ag + g^{-1}dg$ and Goldstone boson goes as $U \to g^{-1}Uh$, both $A^U$ and $A_{\mathfrak{h}}^U$ transform as the connection on $H$ according to Eq. (III.7). The $\mathsf{CS}^{(2n+1)}(A^U, A_{\mathfrak{h}}^U)$ is invariant under the gauge transformation, this can be explicitly checked. But the second term $-\mathsf{CS}^{(2n+1)}(A, A_{\mathfrak{h}})$ part will give the anomaly associated with the symmetry $G$ and $H$ which needs to be canceled by the SPT in the bulk, this mechanism is called anomaly inflow [57], and the bulk boundary combined system is non-anomalous. In short, the gauged WZW term Eq. (IV.3) has the anomaly associated with the bulk SPT which is described by $\mathsf{CS}^{(2n+1)}(A, A_{\mathfrak{h}})$. If we assume there is no anomaly associated to the symmetry $H$, then the gauged WZW term matches with the anomaly of the bulk SPT described by $\mathsf{CS}^{(2n+1)}(A, A_{\mathfrak{h}})$ and $d\mathsf{CS}^{(2n+1)}(A, A_{\mathfrak{h}}) = \mathsf{ch}^{(2n+2)}(A)$.

We summarize the anomaly matching by WZW term in the following, given the WZW term $\Gamma^{(d+1)}(U)$ with field $U$ lives in $G/H$, couple it to the background gauge fields $A, A'$ associated with

global symmetry $G, H$ and get $\Gamma^{(d+1)}(U, A, A')$. Under the gauge transformation, the anomalous phase of the gauged WZW term is canceled by the bulk SPT described by the (relative) Chern-Simons term.

Another way to see the necessity of relative Chern-Simons term is as follows, supposing the UV symmetry $G$ has 't Hooft anomaly and it is matched by $\mathsf{CS}^{(2n+1)}(A)$, the IR symmetry $H$ also has the 't Hooft anomaly and matched by $\mathsf{CS}^{(2n+1)}(A_{\mathfrak{h}})$. The gauged WZW term together with the IR anomaly should match the UV anomaly, in other word, the gauged WZW term should yields the same anomalous phase as $\mathsf{CS}^{(2n+1)}(A) - \mathsf{CS}^{(2n+1)}(A_{\mathfrak{h}})$ which is roughly the relative Chern-Simons term $\mathsf{CS}^{(2n+1)}(A, A_{\mathfrak{h}}) = \mathsf{CS}^{(2n+1)}(A) - \mathsf{CS}^{(2n+1)}(A_{\mathfrak{h}}) + d\beta^{(2n)}(A, A_{\mathfrak{h}})$, where $\beta^{(2n)}(A, A_{\mathfrak{h}})$ is some $2n$-form depending on $A, A_{\mathfrak{h}}$.

In the following, we will use the Cartan homotopy method to find the explicit form of $\mathsf{CS}(A^U, A_{\mathfrak{h}}^U)$, $\alpha^{(d)}(U, A, A_{\mathfrak{h}})$, and compare them with the simple case when $\mathfrak{h} = \varnothing$.

### A. Cartan's homotopy method and relative Chern-Simons term

We postpone the detailed review of Cartan's homotopy method to Appendix. B [58]. For any invariant polynomial $S(A, F)$ of connection $A$ and curvature $F = dA + A \wedge A$, ($dA$ can be substituted by $F - A^2$ and $dF$ is substituted by $-[A, F]$), we have the following formula,

$$(d\ell_t + \ell_t d)S(A_t, F_t) = \delta t \frac{\partial}{\partial t} S(A_t, F_t), \tag{IV.5}$$

where the operator $\ell_t$ is an anti-derivative operator,

$$\ell_t(\eta^{(p)} \wedge \omega^{(q)}) = (\ell_t \eta^{(p)}) \wedge \omega^{(q)} + (-1)^p \eta^{(p)} \wedge (\ell_t \omega^{(q)}). \tag{IV.6}$$

If $A_0, A_1$ both are connections on the same bundle, one can define a one-parameter family, $A_t = A_0 + t(A_1 - A_0)$, and the curvature is given by $F_t = dA_t + A_t \wedge A_t$. The operator $\ell_t$ acts on $A_t, F_t$ as,

$$\ell_t A_t = 0, \quad \ell_t F_t = \delta t(A_1 - A_0). \tag{IV.7}$$

Integrating over $t$ from 0 to 1 on both sides of Eq. (IV.5), we have,

$$S(A_1, F_1) - S(A_0, F_0) = \left(d \int_t \ell_t + \int_t \ell_t d\right) S(A_t, F_t). \tag{IV.8}$$

#### 1. Relative Chern-Simons form

Since the Chern class is even degree closed form $\mathsf{ch}^{(2n)} = \frac{1}{n!}\mathrm{tr}(\frac{\mathrm{i}F}{2\pi})^n$, it can be locally written as an exact form, $\mathsf{ch}^{(2n+2)}(F) = d\mathsf{CS}^{(2n+1)}(A)$. But this is not globally true, if true then the integral

of Chern class on any closed manifold would yield 0. From Eq. (IV.2), the difference of two Chern classes with curvature $F, F'$ can be written as the relative Chern-Simons term. And according to Cartan's homotopy formula,

$$\mathsf{ch}^{(2n+2)}(F) - \mathsf{ch}^{(2n+2)}(F_\mathfrak{h}) = \left( d \int_t \ell_t + \int_t \ell_t d \right) \mathsf{ch}^{(2n+2)}(F_t) = d \int_t \ell_t \mathsf{ch}^{(2n+2)}(F_t) \equiv d\mathsf{CS}^{(2n+1)}(A, A_\mathfrak{h}), \tag{IV.9}$$

where $F_t = dA_t + A_t^2$, and $A_t = A_\mathfrak{h} + tA_\mathfrak{f}$ as discussed around Eq. (III.8). The relative Chern-Simons term is given by,

$$\mathsf{CS}^{(2n+1)}(A, A_\mathfrak{h}) = \int_t \ell_t \mathsf{ch}^{(2n+2)}(F_t) = \frac{1}{n!} \left( \frac{\mathbf{i}}{2\pi} \right)^{n+1} \int dt \ \mathrm{tr}(A_\mathfrak{f} F_t^n). \tag{IV.10}$$

More explicitly, we have

$$\mathsf{CS}^{(2n+1)}(A, A_\mathfrak{h}) = \frac{1}{n!} \left( \frac{\mathbf{i}}{2\pi} \right)^{n+1} \int dt \ \mathrm{tr}(A_\mathfrak{f}(F_\mathfrak{h} + t\mathcal{D}_\mathfrak{h} A_\mathfrak{f} + t^2 A_\mathfrak{f})^n), \tag{IV.11}$$

where $\mathcal{D}_\mathfrak{h}$ is the covariant derivative with respect to $\mathfrak{h}$-connection, $\mathcal{D}_\mathfrak{h} A_\mathfrak{f} = dA_\mathfrak{f} + \{A_\mathfrak{h}, A_\mathfrak{f}\}$. The relative Chern-Simons terms with degrees 3 and 5 are,

$$\mathsf{CS}^{(3)}(A, A_\mathfrak{h}) = -\frac{1}{4\pi^2} \mathrm{tr} \left( A_\mathfrak{f} F_\mathfrak{h} + \frac{1}{2} A_\mathfrak{f} \mathcal{D}_\mathfrak{h} A_\mathfrak{f} + \frac{1}{3} A_\mathfrak{f} A_\mathfrak{f} A_\mathfrak{f} \right), \tag{IV.12}$$

$$\mathsf{CS}^{(5)}(A, A_\mathfrak{h}) = -\frac{\mathbf{i}}{16\pi^3} \mathrm{tr} \left( A_\mathfrak{f} F_\mathfrak{h}^2 + \frac{1}{2} A_\mathfrak{f} \{F_\mathfrak{h}, \mathcal{D}_\mathfrak{h} A_\mathfrak{f}\} + \frac{2}{3} F_\mathfrak{h} A_\mathfrak{f}^3 \right.$$

$$\left. + \frac{1}{3} A_\mathfrak{f}(\mathcal{D}_\mathfrak{h} A_\mathfrak{f})^2 + \frac{1}{2} A_\mathfrak{f}^3 \mathcal{D}_\mathfrak{h} A_\mathfrak{f} + \frac{1}{5} A_\mathfrak{f}^5 \right). \tag{IV.13}$$

Since $A_\mathfrak{f}$, the curvature of $A_\mathfrak{h}$ and the covariant derivative with respect to $A_\mathfrak{h}$ are all transformed under adjoint action of $H$, the relative Chern-Simons term is manifestly invariant under the $H$ transformation as well as $G$ transformation. If $H = \varnothing$, $A_\mathfrak{f}$ is identified with the $G$-connection $A$, the curvature $F_\mathfrak{h} = 0$ and $\mathcal{D}_\mathfrak{h} A_\mathfrak{f} = dA$, then,

$$\mathsf{CS}^{(3)}(A) = -\frac{1}{8\pi^2} \mathrm{tr} \left( AdA + \frac{2}{3} A^3 \right), \tag{IV.14}$$

$$\mathsf{CS}^{(5)}(A) = -\frac{\mathbf{i}}{48\pi^3} \mathrm{tr} \left( A(dA)^2 + \frac{3}{2} A^3 dA + \frac{3}{5} A^5 \right), \tag{IV.15}$$

which reproduce the Chern-Simons forms with only the background field associated with the global symmetry $G$.

### 2. Explicit form of gauged WZW term

According to the definition of gauged WZW term $\underline{\Gamma}^{(d+1)}(U, A, A_\mathfrak{h})$ in Eq. (IV.3), the gauged WZW term is obtained by the difference of relative Chern-Simons terms with connection $A^U$ and

$A$ which result in a closed $d+1$-form $\Gamma^{(d+1)}(U)$ only depending on the Goldstone boson configuration $U$ and an exact $d+1$-form expressed as $d\alpha^{(d)}(U, A)$ which depends on the configuration $U$ as well as the gauge field $A$. We will use Cartan's homotopy formula to obtain the explicit form of the $d$-form $\alpha^{(d)}$ and give the explicit form of the gauged WZW term.

The closed $d+1$-form $\Gamma^{(d+1)}(U)$ is easily obtained by turning off the background gauge field $A$ in the relative Chern-Simons forms Eq. (IV.12) and Eq. (IV.13). The connection $A^U$ defined in Eq. (III.5) and Eq. (III.7) can be decomposed as,

$$A^U = A_{\mathfrak{h}}^U + A_{\mathfrak{f}}^U = (U^{-1}A_{\mathfrak{h}}U + V) + (U^{-1}A_{\mathfrak{f}}U + \phi). \tag{IV.16}$$

Once turning off the background gauge field, $A_{\mathfrak{h}}^U = V, A_{\mathfrak{f}}^U = \phi$ and the curvature of $A_{\mathfrak{h}}$ becomes $W = dV + V^2$, then $\Gamma^{(d+1)}(U)$ becomes,

$$\Gamma^{(3)}(U) = \mathsf{CS}^{(3)}(\theta, V) = -\frac{1}{4\pi^2}\mathrm{tr}\left(\phi W + \frac{1}{2}\phi\mathcal{D}_V\phi + \frac{1}{3}\phi^3\right) \quad \in H^3(G/H, \mathbb{R}), \tag{IV.17}$$

$$\Gamma^{(5)}(U) = \mathsf{CS}^{(5)}(\theta, V) = -\frac{\mathbf{i}}{16\pi^3}\mathrm{tr}\left(\phi W^2 + \frac{1}{2}\phi\{W, \mathcal{D}_V\phi\} + \frac{2}{3}W\phi^3\right.$$

$$\left. + \frac{1}{3}\phi(\mathcal{D}_V\phi)^2 + \frac{1}{2}\phi^3\mathcal{D}_V\phi + \frac{1}{5}\phi^5\right) \quad \in H^5(G/H, \mathbb{R}), \tag{IV.18}$$

where $\mathcal{D}_V\phi = d\phi + \{V, \phi\} = 0$. One can check when $H = \varnothing$, $\phi$ is identified with $\theta$, the curvature $W$ vanishes and the covariant derivative $\mathcal{D}_V\phi = d\theta = -\theta \wedge \theta$, the WZW terms become,

$$\Gamma_G^{(3)}(U) = -\frac{1}{24\pi^3}\mathrm{tr}\theta^3 = -\frac{1}{24\pi^3}\mathrm{tr}(U^{-1}dU)^3 \quad \in H^3(G, \mathbb{R}), \tag{IV.19}$$

$$\Gamma_G^{(5)}(U) = -\frac{\mathbf{i}}{480\pi^3}\mathrm{tr}\theta^5 = -\frac{\mathbf{i}}{480\pi^3}\mathrm{tr}(U^{-1}dU)^5 \quad \in H^5(G, \mathbb{R}). \tag{IV.20}$$

These match with the standard WZW term for WZW conformal field theory in 2d and chiral symmetry breaking in 4d. The WZW terms for $G$ and $G/H$ obtained by the Cartan homotopy method reproduce those in Sec. II.

The gauged WZW defined in Eq. (IV.3) can be obtained by considering the interpolation $A_t = tU^{-1}AU + \theta, A_{\mathfrak{h},t} = tU^{-1}A_{\mathfrak{h}}U + V$, in this case the Cartan homotopy formula becomes,

$$\mathsf{CS}^{(2n+1)}(A^U, A_{\mathfrak{h}}^U) - \mathsf{CS}^{(2n+1)}(\theta, V) = \left(d\int_t \ell_t + \int_t \ell_t d\right)\mathsf{CS}^{(2n+1)}(A_t, A_{\mathfrak{h},t})$$

$$= d\alpha^{(2n)} + \int_t \ell_t(\mathsf{ch}^{(2n+2)}(F_t) - \mathsf{ch}^{(2n+2)}(F_{t\mathfrak{h}})) = d\beta^{(2n)} + \mathsf{CS}^{(2n+1)}(A) - \mathsf{CS}^{(2n+1)}(A_{\mathfrak{h}}). \tag{IV.21}$$

Since there is no anomaly for $H$, the Chern-Simons term for gauge field $A_{\mathfrak{h}}$ vanishes. The Chern-Simons term $\mathsf{CS}^{(2n+1)}(A)$ differs from the relative Chern-Simons term $\mathsf{CS}^{(2n+1)}(A, A_{\mathfrak{h}})$ by a total derivative $d\beta^{(2n)}$, therefore, the gauged WZW term is,

$$\underline{\Gamma}^{(2n+1)}(U, A) \equiv \mathsf{CS}^{(2n+1)}(A^U, A_{\mathfrak{h}}^U) - \mathsf{CS}^{(2n+1)}(A, A_{\mathfrak{h}}) = \Gamma^{(2n+1)}(U) + d(\alpha^{(2n)} + \beta^{(2n)}), \tag{IV.22}$$

where the $2n$-form $\alpha$ depends on $U, A, A_\mathfrak{h}$ while $\beta$ depends only on the background gauge fields. For example, $\beta^{(2)} = \mathrm{tr}(A \wedge A_\mathfrak{h})$, $\alpha^{(2)} = \mathrm{tr}(\phi U^{-1}(A + A_\mathfrak{h})U)$. If $H = \varnothing$, $\beta^{(2n)}$ vanishes, and $\alpha^{(2n)}$ is obtained by,

$$\int_t \ell_t \mathrm{tr}(\mathsf{CS}^{(2n+1)}(tA + dUU^{-1})). \tag{IV.23}$$

This gives, for example, $\alpha^{(2)} = \frac{1}{2}\mathrm{tr}(dUU^{-1}A)$. More details can be found in Appendix. B.

## V.   NONLINEAR SIGMA MODEL OF DQCP THEORIES

With the gears presented in the previous sections, we will show that the WZW term captures the intertwinement of the topological defects and matches with the 't Hooft anomaly in the anomalous theory. We use the charge operators of topological defects to construct the WZW term and use the gauged WZW term to match the mixed anomaly in the theory.

We are interested in the critical points or phases between spontaneously symmetry-breaking phases in the presence of the 't Hooft anomaly, and the manifestation of the 't Hooft anomaly in different symmetry-breaking phases. This situation is along with many deconfined quantum critical point theories. We begin by revisiting the intertwinement in the DQCP theory of the 3d quantum magnet [10, 12], and then construct a series of 4d DQCP theories which is motivated by recent work on DQCP among grand unified theories [25, 26]. The new set of 4d deconfined quantum criticality theories is the higher dimensional generalization of 3d DQCP theories, and the new 4d DQCP (DQCPh) theories are governed by 't Hooft anomaly of global $\mathsf{SO}(2n)$ symmetry which turns out to be a variation of the new $\mathsf{SU}(2)$ anomaly [25, 26, 28].

### A.   Revisiting deconfined quantum critical point in 3d quantum magnet

As mentioned in the introduction, the continuous global symmetry of the 3d quantum magnet is $G = \mathsf{SO}(3)_S \times \mathsf{SO}(2)_R$, which corresponds to spin and lattice rotation symmetry. The Néel phase in this system is the antiferromagnetic phase, with spins pointing up or down. This phase has $H_{\mathrm{Néel}} = \mathsf{SO}(2)_S \times \mathsf{SO}(2)_R$ symmetry, broken by the easy-axis spin configuration. The Goldstone boson in the Néel phase lives in the coset $G/H_{\mathrm{Néel}} \cong \mathsf{SO}(3)_S/\mathsf{SO}(2)_S \cong \mathbb{S}^2_S$. The possible topological defect is classified by $\pi_2(\mathbb{S}^2_S) = \mathbb{Z}$, corresponding to codimension 3 integer-valued defect, which is the hedgehog defect in the Néel phase. On the other hand, the lattice rotation symmetry is broken in the VBS phase, $H_{\mathrm{VBS}} = \mathsf{SO}(3)_S$. The corresponding Goldstone boson lives in the coset

$G/H_{\text{VBS}} = \mathsf{SO}(2)_R \cong \mathbb{S}^1_R$. The possible topological defect is classified by $\pi_1(\mathbb{S}^1_R) = \mathbb{Z}$, which corresponds to codimension 2 integer-valued defects, this is the vortex line in the VBS phase.

It is very interesting that the vortex core in the VBS phase carries spin-$\frac{1}{2}$ degree of freedom [5]. When proliferating the vortices in the VBS phase, the defects will destroy the VBS ordered phase, but due to the additional spin-$\frac{1}{2}$ degree of freedom at the vortex core, the system will become the ordered Néel phase. In other words, the disorder operator of lattice rotation symmetry carries the symmetry charge of the spin rotation symmetry, which is reminiscent of the mixed 't Hooft anomaly of these two symmetries.

The 't Hooft anomaly should match along the renormalization group flow, meaning that all possible phases should have such an anomaly. In the ordered phases, although some defects may be suppressed by energy, their intertwined feature should manifest thanks to the anomaly. We can deform the theory by tuning the relevant operators such that the theory flow to the IR phase with the smallest symmetry $K$ and there is no anomaly with $K$, hence, the anomaly in the UV is matched by the Goldstone boson in the coset $G/K$.

Theories with the same symmetry properties and anomaly will be dual to each other, in the sense that they describe the same IR phase with different UV details [59]. We consider the deformation of the theory to the spontaneous symmetry breaking phase with unbroken symmetry $K = H_{\text{Néel}} \cap H_{\text{VBS}} = \mathsf{SO}(2)_S$. Then the gapless theory of the Goldstone boson living in the coset $G/K = \mathbb{S}^2_S \times \mathbb{S}^1_R$ with WZW term would be a suitable dual description of the DQCP theory. Indeed, we will show this construction is related to the $\mathsf{O}(5)$ nonlinear sigma model that served as the dual theory of various gauge theory descriptions of 3d DQCP [12]. Both topological defects in Néel and VBS phase are present in this Goldstone boson theory, since $\pi_k(\mathbb{S}^2_S \times \mathbb{S}^1_R) = \pi_k(\mathbb{S}^2_S) \times \pi_k(\mathbb{S}^1_R)$. The symmetry-breaking routes are summarized as follows,

$$G = \mathsf{SO}(3)_S \times \mathsf{SO}(2)_R$$

$$\begin{array}{ccc} & \xrightarrow{\langle\Phi_N\rangle\neq 0} & \xrightarrow{\langle\Phi_V\rangle\neq 0} \\ H_{\text{Néel}} = \mathsf{SO}(2)_S \times \mathsf{SO}(2)_R & & H_{\text{VBS}} = \mathsf{SO}(3)_S \quad (\text{V.1}) \\ & \xrightarrow{\langle\Phi_V\rangle\neq 0} \quad \xrightarrow{\langle\Phi_N\rangle\neq 0} \\ & K = H_{\text{Néel}} \cap H_{\text{VBS}} = \mathsf{SO}(2)_S \end{array}$$

We denote the generators of $\mathsf{SO}(3)_S$ as $\{T^1, T^2, T^3\}$ and $\mathsf{SO}(2)_R$ as $\{T^4\}$. Supposing the generators of $H_{\text{Néel}}$ are $\{T^3, T^4\}$, then the charge operators of the topological defects in the Néel and VBS phase are represented by,

$$\tilde{\eta}^{(1)} = \theta^4 \in H^1(G/H_{\text{VBS}}, \mathbb{R}), \quad \tilde{\xi}^{(2)} = \theta^1 \wedge \theta^2 \in H^2(G/H_{\text{Néel}}, \mathbb{R}). \tag{V.2}$$

In the ordered phase with global symmetry $K = H_{\text{Néel}} \cap H_{\text{VBS}} = \mathsf{SO}(2)_S$, the generator is $\{T^3\}$, and the cohomology generators are,

$$\eta^{(1)} = \theta^4 \in H^1(G/K, \mathbb{R}), \quad \xi^{(2)} = \theta^1 \wedge \theta^2 \in H^2(G/K, \mathbb{R}). \tag{V.3}$$

In this case, $\tilde{\eta}^{(1)} = \eta^{(1)}$ and $\tilde{\xi}^{(2)} = \xi^{(2)}$ since the global symmetry $G$ is the tensor product of two subgroups. Wedge product of the two generators yields $a$ generator of the higher degree cohomology group, $\eta^{(1)} \wedge \xi^{(2)} = \theta^4 \wedge \theta^1 \wedge \theta^2 \in H^3(G/K, \mathbb{R})$.

The WZW term in the 2+1d DQCP assigns a phase to the linking between the VBS vortex and hedgehog defect or the linking between $\mathbb{S}^2_S$ and $\mathbb{S}^1_R$ in $\mathbb{S}^4$ [60]. The way to define the linking is to find the surface $D^2_R$ that is bounded by the circle $\mathbb{S}^1_R$ and the intersection with $\mathbb{S}^2_S$ gives the linking number. The form on $D^2_R$ is denoted by $\hat{\eta}^{(1)}$ and the WZW term is,

$$\Gamma^{(4)} = \xi^{(2)} \wedge d\hat{\eta}^{(1)}. \tag{V.4}$$

More explicitly, we can parameterize $\mathbb{S}^2_S$ and $\mathbb{S}^1_R$ by 3-component unit-vector $\boldsymbol{n}_S$ and 2-component unit-vector $\boldsymbol{n}_R$, then $\xi^{(2)} = \epsilon_{abc} n^a_S dn^b_S \wedge dn^c_S$, $\eta^{(1)} = n^1_R dn^2_R$. The 1-form is closed when restricting on the circle, but not closed on the disk, $d\hat{\eta}^{(1)} = dn^1_R dn^2_R$. Therefore, the WZW term is given by $\Gamma^{(4)} = \epsilon_{abcde} n^a_S dn^b_S dn^c_S dn^d_R dn^e_R$, which appears in the $\mathsf{SO}(5)$ NLSM of DQCP [12, 61–64].

When coupled to the background gauge field the anomaly of the gauged WZW term comes from the mixed $\theta$ term in 4d which matches the anomaly in the bulk SPT phase with global symmetry $\mathsf{SO}(3)_S \times \mathsf{SO}(2)_R$. We can further embed $\mathsf{SO}(3)_S \times \mathsf{SO}(2)_R \hookrightarrow \mathsf{SO}(5)$, the corresponding anomaly is described by $\frac{1}{2} w_4 \in H^4(BSO(5), U(1))$, upon pull-back to $\mathsf{SO}(3)_S \times \mathsf{SO}(2)_R$, the anomaly becomes,

$$\frac{1}{2} w_4(A_S \oplus A_R) = \frac{1}{2} w_2(A_S) w_2(A_R) = \frac{1}{2} \frac{F_R}{2\pi} w_2(A_S), \tag{V.5}$$

where $F_R = dA_R$ and $w_2$ is the second Stiefel Whitney class of $SO(3)$ bundle. This anomaly also matches with that in 3d $\mathbb{C}P^1$ model in [10, 11]. The same anomaly in WZW theory and 3d $\mathbb{C}P^1$ model is also a check of the infrared duality of the different theories [12, 59]. Recent works on quantum spin liquid have examined related gauge theories and their corresponding NLSM with WZW term, and the target spaces of the NLSM are Stiefel manifold or Grassmannian manifold [22–24]. It would also be interesting to construct other 2+1d DQCP theories that saturate the anomaly discussed in [65, 66]. Similar construction and anomaly matching can be applied to 1+1d system [7, 67–70].

**B. Deconfined quantum critical point and intertwinement of topological defects in 4d**

In contrast to the extensive theoretical and numerical studies of deconfined quantum critical points in 2d and 3d, the 4d generalization of DQCP is rarely explored. One difficulty is that the gauge fields tend to be deconfined in higher dimensions and many gauge theories then describe the deconfined quantum critical phases instead of critical points. Nevertheless, previous works focus on the gauge theory description and find interesting examples of deconfined quantum critical point with mixed 't Hooft anomaly that implies the intertwinement of symmetries [71].

Regardless of the specific models and their critical behaviors, it is interesting to study the intertwinement of topological defects in 4d which manifests the 't Hooft anomaly in the UV theory, since there are more types of topological defects in higher dimensions. This also offers a way to understand the higher dimensional SPTs, since the WZW term essentially describes the linking between extended operators in the bulk where the SPT lives in.

We construct the nonlinear sigma model with WZW term to describe this phenomenon in the following subsections and construct the corresponding fermionic parton theories in Sec. VI. Our construction turns out to describe the deconfined quantum critical phase (DQCPh), since the minimal fermionic parton theory contains $U(1)$ gauge field, which is deconfined in the 3+1d [25, 26]. Because the inputs of our construction are the global symmetries and corresponding mixed 't Hooft anomaly, it may have different gauge theory descriptions with the same global symmetry and anomaly. It is interesting to find specific gauge theory description that realizes the deconfined quantum critical point.

From the previous discussion, the DQCP theory is anomalous and can be thought of as the boundary of one higher dimensional SPT. The 4d DQCPh theory can serve as the boundary of 5d SPT. The 5d SPT with only $\mathbb{Z}_2^T$ symmetry is $\mathbb{Z}_2$ classified, this SPT is described by $\int_{Y^5} w_2 w_3$ [72, 73] and recently studied in [74–76], where $w_i \in H^i(Y^5, \mathbb{Z}_2)$ is the $i^{\text{th}}$ Stiefel-Whitney class. In the presence of symmetry, similar topological terms are possible for the all-fermion electrodynamics [77] and the new $SU(2)$ anomaly [28]. Hence, it is possible to have an anomalous theory at the boundary of the nontrivial SPT with the anomaly described in the above mentioned examples. In the following, we will discuss a series of models with the new $SU(2)$ anomaly, these models describe the gapless theories between two spontaneously symmetry-breaking phases.

### 1. Symmetry breaking and topological defects

Recent work shows that the DQCP (or DQCPh, depending on the model details) can present among the Grand Unified Theories (GUTs) in which the standard model with global symmetry generated by $\mathfrak{k}_{\mathrm{SM}} = \mathfrak{su}(3) \oplus \mathfrak{su}(2)_L \oplus \mathfrak{u}(1)_Y \oplus \mathfrak{u}(1)_X$ can be embedded, the three GUTs are $\mathsf{SO}(10)$ GUT with $\mathfrak{g}_{\mathsf{SO}(10)} = \mathfrak{so}(10)$, Pati-Salam (PS) model with $\mathfrak{h}_{\mathrm{PS}} = \mathfrak{so}(6) \oplus \mathfrak{so}(4) = \mathfrak{su}(4) \oplus \mathfrak{su}(2)_L \oplus \mathfrak{su}(2)_R$ and Georgi-Glashow (GG) model with $\mathfrak{h}_{\mathrm{GG}} = \mathfrak{su}(5) \oplus \mathfrak{u}(1)_X$ [25, 26]. The gauge symmetry is "ungauging" such that they can be viewed as global symmetries. In other words, the dynamical gauge fields in the original theories are background gauge fields in these alternative theories. Therefore, the Higgs phases and transitions become symmetry-breaking phases and transitions. When condensing the symmetric $\Phi_{\mathbf{54}}$ or antisymmetric $\Phi_{\mathbf{45}}$ scalar fields charged under $\mathsf{SO}(10)$ in the $\mathsf{SO}(10)$ GUT described in [25], one can get the symmetry breaking phases with unbroken symmetry $\mathfrak{h}_{\mathrm{PS}}$ or $\mathfrak{h}_{\mathrm{GG}}$ respectively, these symmetries can be further broken down to $\mathfrak{k}_{SM}$. The symmetry-breaking pattern is summarized as follows.

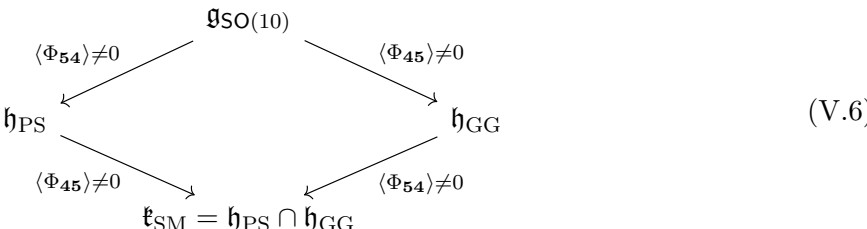

$$\text{(V.6)}$$

where $\Phi_{\mathbf{45}}, \Phi_{\mathbf{54}}$ are traceless symmetric and antisymmetric higgs fields of $\mathsf{SO}(10)$. The possible topological defects in the PS and GG phases are classified by,

$$\pi_2 \left( \frac{\mathsf{SO}(10)}{\mathsf{SO}(6) \times \mathsf{SO}(4)} \right) = \mathbb{Z}_2, \quad \pi_2 \left( \frac{\mathsf{SO}(10)}{\mathsf{U}(5)} \right) = \mathbb{Z}. \tag{V.7}$$

The topological defects in the symmetry breaking phases are codimension 3 defects corresponding to the line operators in 4d. The topological defects in the PS phase are Grassmannian manifold and $\mathbb{Z}_2$ classified, meaning that two of such defects can be deformed to nothing.

As discussed around Eq. (II.4) and Sec. V A, since the coset $\mathsf{SO}(2n)/(\mathsf{SO}(2m) \times \mathsf{SO}(2n-2m))$ and $\mathsf{SO}(2n)/\mathsf{U}(n)$ have vanishing $\pi_1$ and $\pi_0$, the 2nd homotopy group is isomorphic to the 2nd homology group. We can use the corresponding cohomology generators as the charge operators of these topological defects. However, it is impossible to directly express the charge operator of $\mathbb{Z}_2$ classified topological defect of $\mathsf{SO}(2n)/(\mathsf{SO}(2m) \times \mathsf{SO}(2n-2m))$ within the de Rham cohomology, more generally it is impossible for the $\mathbb{Z}_n$ classified topological defects, since the normalized generator of de Rham cohomology yields $\mathbb{Z}$-valued closed form. Mathematically, one may consider $\mod n$

reduction of the cohomology group or using the Cech cohomology. But the current situation is reminiscent of the non-perturbative $\mathsf{SU}(2)$ anomaly which is characterized by $\pi_4(\mathsf{SU}(2)) = \mathbb{Z}_2$ [27], the way to reproduce the non-perturbative anomaly perturbatively is by embedding $\mathsf{SU}(2) \hookrightarrow \mathsf{SU}(3)$, and the non-perturbative $\mathsf{SU}(2)$ anomaly is seen by the WZW term of $\mathsf{SU}(3)$ group [21]. As in Sec. V A, we attempt to embed the space $\mathsf{SO}(2n)/(\mathsf{SO}(2m) \times \mathsf{SO}(2n-2m))$ into a larger space, the natural choice is the Goldstone boson in the SM phase with both order parameters condensed and the unbroken symmetry is $K$. Indeed, we find that the embedding into $G/K$ can capture both topological defects even this $\mathbb{Z}_2$ classified topological defects in the PS phase with target space $\mathsf{SO}(2n)/(\mathsf{SO}(2m) \times \mathsf{SO}(2n-2m))$.

The above statement can be seen by examining the homotopy group of the target space $G/K$. The short exact sequence of the global symmetry $G, K$ and coset is $0 \to K \to G \to G/K \to 0$, this induces the long exact sequence for the homotopy group, and the relevant part is,

$$... \to \pi_2(G) \to \pi_2(G/K) \to \pi_1(K) \to \pi_1(G) \to \pi_1(G/K) \to ... \tag{V.8}$$

The homotopy groups of $G = \mathsf{SO}(n)$ is known, $\pi_2(\mathsf{SO}(n)) = 0, \pi_1(\mathsf{SO}(n)) = \mathbb{Z}_2, n \geq 3$ and $G/K$ is contractible $\pi_1(G/K) = 0$, the long exact sequence becomes,

$$0 \to \pi_2(G/K) \to \pi_1(K) \to \mathbb{Z}_2 \to 0 \tag{V.9}$$

Therefore $\pi_2(G/K) = \pi_1(K) = \pi_1(\mathsf{SU}(3) \times \mathsf{SU}(2) \times \mathsf{U}(1) \times \mathsf{U}(1)) = \mathbb{Z} \oplus \mathbb{Z}$ quotient by $\mathbb{Z}_2$, where the two $\mathbb{Z}$s correspond to the topological defects in PS and GG phase respectively. This construction is valid for a series of theories with $G = \mathsf{SO}(2n), n \geq 2$.

### 2.  Construction of Lie algebras

In this subsection, we describe the embedding of $\mathfrak{so}(2m) \oplus \mathfrak{so}(2n-2m)$ and $\mathfrak{u}(n)$ into the $\mathfrak{so}(2n)$ Lie algebra. The $\mathfrak{so}(2n)$ Lie algebra is represented by $n(2n-1)$ $2n \times 2n$ anitsymmetric real matrices, which generate the rotation of a $2n$-vector. $\mathfrak{so}(2m) \oplus \mathfrak{so}(2n-2m)$ consists of the $2n \times 2n$ anitsymmetric real matrices that rotate within the first $2m$ elements, or within the last $2(n-m)$ elements in the $2n$-vector. The $\mathfrak{u}(n)$ is embedded in the $\mathfrak{so}(2n)$ by Kronecker producting the symmetric generators of $\mathfrak{u}(n)$ with $\mathbf{i}\sigma^2$ and antisymmetric generators of $\mathfrak{u}(n)$ with $\mathbf{i}\sigma^0$.

In our case, $m = 2\lfloor n/2 \rfloor$ in $\mathfrak{so}(2m) \oplus \mathfrak{so}(2n-2m)$, where $m$ is taking the floor of $n/2$. The intersection of $\mathfrak{u}(n)$ and $\mathfrak{so}(2m) \oplus \mathfrak{so}(2n-2m)$ is isomorphic to $\mathfrak{u}(m) \oplus \mathfrak{u}(n-m)$ and always contain two $\mathfrak{u}(1)$s, in the upper-left block $\mathfrak{u}(1)_+$ and lower-right block $\mathfrak{u}(1)_-$ of the original $\mathfrak{so}(2n)$

respectively. Hence, $\mathfrak{u}(1)_+ + \mathfrak{u}(1)_- \subset \mathfrak{u}(n)$ rotates the upper and lower block with the same phase, while $\mathfrak{u}(1)_+ - \mathfrak{u}(1)_- \subset \mathfrak{so}(2m) \oplus \mathfrak{so}(2n - 2m)$ rotates the upper and lower block with the opposite phase.

Since the intersection of Lie algebras is $\mathfrak{u}(m) \oplus \mathfrak{u}(n-m)$, the symmetry $K$ generated by this Lie algebra contains two $\mathsf{U}(1)$ factors, $\pi_2(G/K) = \pi_1(K) = \mathbb{Z} \oplus \mathbb{Z}$ due to Eq. (V.9). We have identified that one of the $\mathsf{U}(1)$ factors is in $\mathsf{U}(n)$, the other relates to $\mathsf{SO}(2m) \times \mathsf{SO}(2n - 2m)$. Hence, the topological defects in $G/K$ correspond to those in symmetry breaking phases with only unbroken $\mathsf{U}(n)$ or $\mathsf{SO}(2m) \times \mathsf{SO}(2n - 2m)$. Since they are $\mathbb{Z}$ classified, we can find the de Rham cohomology expressions of the charge operators corresponding to the topological defects,

$$\eta^{(2)} \in H^2(G/K, \mathbb{R}), \quad \xi^{(2)} \in H^2(G/K, \mathbb{R}), \tag{V.10}$$

where $\eta^{(2)} = \tilde{\eta}^{(2)} \in H^2(G/U(n), \mathbb{R})$ corresponds to the charge operator of topological defect in $G$ breaking down to $H_{\mathsf{U}} = \mathsf{U}(n)$ phase, and $\xi^{(2)}$ should relate to $\tilde{\xi}^{(2)} \in H^2(G/\mathsf{SO}(2m)\mathsf{SO}(2n - 2m), \mathbb{Z}_2)$ which corresponds to the charge operator of topological defect in $G$ breaks down to $H_{\mathsf{SO}} = \mathsf{SO}(2m) \times \mathsf{SO}(2n - 2m)$ phase.

For $\mathsf{SO}(8)$, these generators of the cohomology group is given by,

$$\eta_{\mathsf{U}}^{(2)} = \theta^1 \wedge \theta^7 + \theta^6 \wedge \theta^{12} + \theta^2 \wedge \theta^8 + \theta^3 \wedge \theta^9 + \theta^4 \wedge \theta^{10} + \theta^5 \wedge \theta^{11} \in H^2(G/K, \mathbb{R}), \tag{V.11}$$

$$\xi_{\mathsf{SO}}^{(2)} = -\theta^1 \wedge \theta^7 + \theta^6 \wedge \theta^{12} + \theta^{14} \wedge \theta^{20} + \theta^{15} \wedge \theta^{21} + \theta^{16} \wedge \theta^{22} + \theta^{17} \wedge \theta^{23} \in H^2(G/K, \mathbb{R}), \tag{V.12}$$

where the indices $1 \sim 12$ are the generators of the coset $\mathfrak{so}(8)/\mathfrak{u}(4)$, while $\{2, 3, 4, 5, 8, 9, 10, 11, 14, 15, 16, 17, 20, 21, 22, 23\}$ are the generators of the coset $\mathfrak{so}(8)/\mathfrak{so}(4) \oplus \mathfrak{so}(4)$. The $\eta_{\mathsf{U}}^{(2)} \in H^2(G/K, \mathbb{R})$ coincides with the nontrivial generator in $H^2(\mathsf{SO}(8)/\mathsf{U}(4), \mathbb{R})$. And it is worth mentioning that the first two terms in $\xi_{\mathsf{SO}}^{(2)}$ will cancel each other when pull-back to $\mathbb{S}^2$, the remain terms are all in $\mathfrak{so}(8)/\mathfrak{so}(4) \oplus \mathfrak{so}(4)$, this further supports that $\xi_{\mathsf{SO}}^{(2)}$ relates to the generator in $H^2(\mathsf{SO}(8)/\mathsf{SO}(4)\mathsf{SO}(4), \mathbb{Z}_2)$.

### 3. Wess-Zumino-Witten term

As illustrated in Sec. V A, one can construct $a$ WZW term by wedge product of the charge operators,

$$\eta_{\mathsf{U}}^{(2)} \wedge \xi_{\mathsf{SO}}^{(2)} \in H^4(G/K, \mathbb{R}). \tag{V.13}$$

However, to properly include the linking information, an additional degree of freedom should be included. Intuitively, two 2-spheres can link with each other in $\mathbb{S}^5$ but cannot be properly described

in 4-dimension, this is similar to the lower dimension example that the linking of two circles needs to be embedded in $\mathbb{S}^3$ and the linking is essentially the intersection between one circle and the disk that is bounded by the other circle. Following this procedure, one needs to find the 3-disk $D^3$ that is bounded by one of the 2-spheres, say corresponding to $\xi_{\mathsf{SO}}^{(2)}$, then $\xi_{\mathsf{SO}}^{(2)}$ is no longer closed, and the WZW term that encodes the linking of two topological defects is,

$$\Gamma^{(5)} = \eta_{\mathsf{U}}^{(2)} \wedge d\xi_{\mathsf{SO}}^{(2)} \in H^5(\widehat{G/K}, \mathbb{R}), \tag{V.14}$$

where $\widehat{G/K}$ is the extension of $G/K$ such that it contains a 3-disk which is bounded by a 2-sphere. This term corresponds to $\phi W_1 W_2$ in Eq. (IV.18). As mentioned in [26], the mixed anomaly of $\mathsf{U}(n)$ and $\mathsf{SO}(2m) \times \mathsf{SO}(2n-2m)$ is contained in $\mathsf{SO}(2n)$. The gauged WZW term then matches with the anomaly from the Chern-Simons term, for the $\mathsf{SO}(2n)$ global symmetry, $d\mathsf{CS}^{(5)} = W_3(\mathsf{SO}(2n))^2 \in H^6(BSO(2n), \mathbb{Z})$, whose mod 2 reduction is $w_3(\mathsf{SO}(2n))^2 \in H^6(BSO(2n), \mathbb{Z}_2)$ corresponding to the image of $w_2 w_3(\mathsf{SO}(2n))$ [31]. This model is akin to the Stiefel liquid in 2+1d, where the target space is Stiefel manifold, and the anomaly is carefully studied in [22, 23].

### 4. Intertwinement of the topological defects and higher linking number

Since both topological defects are codimension 3 and $\mathbb{Z}$ classified, their charge operators are represented by the generators of the second cohomology of $G/K$, $H^2(G/K, \mathbb{Z}) = \mathbb{Z} \oplus \mathbb{Z}$. As discussed previously, the two $\mathbb{Z}$s correspond to the two $\mathfrak{u}(1)$ factors in $K$ and one is in the $\mathfrak{u}(n)$, another is in the $\mathfrak{so}(2m) \oplus \mathfrak{so}(2n-2m)$. To illustrate the intertwinement of the topological defects, we consider two 2-spheres embedded in the target space $G/K$,

$$\mathbb{S}_{\mathsf{U}}^2 \sqcup \mathbb{S}_{\mathsf{SO}}^2 \xrightarrow{f} G/K. \tag{V.15}$$

where $\sqcup$ is the disjoint union of two manifolds. Intuitively, we are considering the mapping that sends two disjoint 2-spheres into the homogeneous space $G/K$, such that the second cohomology of $G/K$ is pull-back via $f^*$ to the second cohomology of each sphere, $f^* \eta_{\mathsf{U}}^{(2)} = \omega_{\mathsf{U}}^{(2)} \in H^2(\mathbb{S}_{\mathsf{U}}^2, \mathbb{R})$ and $f^* \xi_{\mathsf{SO}}^{(2)} = \omega^{(2)}\mathsf{SO} \in H^2(\mathbb{S}_{\mathsf{SO}}^2, \mathbb{R})$. The two different topological defects are then simply understood by these two spheres. The linking of the two spheres is characterized by the degree of the map that sends the disjoint spheres into $\mathbb{S}^5$ [78]. We can further embed $\mathbb{S}^5 \xrightarrow{h} \widehat{G/K}$, then the map is summarized as,

$$\mathbb{S}_{\mathsf{U}}^2 \sqcup \mathbb{S}_{\mathsf{SO}}^2 \xrightarrow{g} \mathbb{S}^5 \xrightarrow{h} \widehat{G/K}. \tag{V.16}$$

Hence, the WZW term $\Gamma^{(5)}$ in Eq. (V.14) is pull-back via $h^*$ to the non-trivial element of $H^5(\mathbb{S}^5, \mathbb{R})$, and $\deg(g) = -\mathsf{Lk}(\mathbb{S}^2_\mathsf{U}, \mathbb{S}^2_\mathsf{SO})$ is the linking number of $\mathbb{S}^2_\mathsf{U}$ and $\mathbb{S}^2_\mathsf{SO}$ in $\mathbb{S}^5$ [78]. The WZW term on $\mathbb{S}^5$ captures the essential intertwinement of the different topological defects in $G/K$.

### 5. Explicit construction for $G = \mathsf{SO}(2n), n \geq 4$

In the following example, we are focusing on global symmetry $\mathsf{SO}(8)$, and the subgroups are $H_\mathsf{SO} = \mathsf{SO}(4) \times \mathsf{SO}(4)$ and $H_\mathsf{U} = \mathsf{U}(4)$. This construction also applies to $G = \mathsf{SO}(2n), n \geq 4$.

We first construct the map from $\mathbb{S}^2_\mathsf{U} \sqcup \mathbb{S}^2_\mathsf{SO} \to G/K$, the two spheres are related to the two $\mathfrak{u}(1)$ factors in $K$. The 2-sphere can be viewed as the homogeneous space $\mathbb{S}^2 = \frac{\mathsf{SO}(3)}{\mathsf{SO}(2)}$, thus, we take the generators of $\mathsf{SO}(3) \subset G$ and modulo the subgroup $\mathsf{SO}(2) \subset G$. Since the two $\mathbb{S}^2$s do not intersect with each other, we take two commuting $\mathfrak{so}(3)$ to construct $\mathbb{S}^2$s, the starting point is,

$$T^A = \{\sigma^{20}, \sigma^{12}, \sigma^{32}\}, \quad \tilde{T}^A = \{\sigma^{02}, \sigma^{21}, \sigma^{23}\}. \tag{V.17}$$

We choose one element of each set as the generator of $\mathsf{SO}(2)$, then the Goldstone boson field for each $\mathbb{S}^2$ is given by, for example,

$$(\theta_1, \phi_1) \in \mathbb{S}^2_\mathsf{U} \to U_1 = e^{\mathbf{i}(\theta_1 \sin\phi_1, \theta_1 \cos\phi_1)\cdot(T^1, T^2)^\intercal} \in \mathsf{SO}(3)/\mathsf{SO}(2)$$

$$(\theta_2, \phi_2) \in \mathbb{S}^2_\mathsf{SO} \to U_2 = e^{\mathbf{i}(\theta_2 \sin\phi_2, \theta_2 \cos\phi_2)\cdot(\tilde{T}^1, \tilde{T}^2)^\intercal} \in \mathsf{SO}(3)/\mathsf{SO}(2) \tag{V.18}$$

It is easy to show that $(U_i dU_i)^2$ corresponds to the generator of $H^2(\mathbb{S}^2_i, \mathbb{R})$. The Goldstone boson fields are equivalent under the right multiplication of $h \in H$, therefore, we have

$$U_1 = \mathbf{n}_1 \cdot (T^A)^\intercal, \quad U_2 = \mathbf{n}_2 \cdot (\tilde{T}^A)^\intercal \tag{V.19}$$

where $\mathbf{n}_i = (\sin\theta_i \cos\phi_i, \sin\theta_i \sin\phi_i, \cos\theta_i)$ is the 3-vector on the 2-sphere, and generator of the second cohomology is given by $\det(\mathbf{n}_i, d\mathbf{n}_i, d\mathbf{n}_i)$. Moreover, one can construct the 6-vector on 5-sphere by interpolating the $\mathbf{n}_i$ vector, the 6-vector is given by $\mathbf{n} = (\cos\psi\mathbf{n}_1, \sin\psi\mathbf{n}_2), \psi \in (0, \pi]$. The corresponding Goldstone boson field is $\cos\psi(\sigma^1 \otimes U_1) + \sin\psi(\sigma^3 \otimes U_2)$. The 5-form is given by $\omega^{(5)} = \det(\mathbf{n}, d\mathbf{n}, d\mathbf{n}, d\mathbf{n}, d\mathbf{n}, d\mathbf{n})$ which is the volume form of $\mathbb{S}^5$, when pull back to $\mathbb{S}^2_\mathsf{U} \sqcup \mathbb{S}^2_\mathsf{SO}$, the integral on $\mathbb{S}^5$ gives the linking number of $\mathbb{S}^2_\mathsf{U}$ and $\mathbb{S}^2_\mathsf{SO}$ in $\mathbb{S}^5$ [78].

The skeleton theory of the Eq. (V.14) together with the kinetic term is given by the $\mathsf{O}(6)$ nonlinear sigma model with WZW term,

$$\int_X \frac{1}{2g}(\partial_\mu \mathbf{n})^2 + \frac{2\pi\mathbf{i}}{\Omega_5} \int_Y \epsilon_{abcdef} n^a dn^b \wedge dn^c \wedge dn^d \wedge dn^e \wedge dn^f \tag{V.20}$$

where $\Omega_5 = \pi^3$ is the area of 5-sphere. Similar action appears in Ref. 79 and previously in Ref. 80. The Eq. (V.20) can be viewed as the boundary of the bulk SPT with $\mathsf{SO}(6)$ symmetry which is described by the nonlinear sigma model with $\theta$-term [81, 82]. With this skeleton theory Eq. (V.20), one can see that the charge operators of the topological defects are given by $\omega_\mathsf{U}^{(2)} = \epsilon_{abc} n^a dn^b \wedge dn^c$ and $\omega_\mathsf{SO}^{(2)} = \epsilon_{def} n^d dn^e \wedge dn^f$ where $a, b, c$ are in $\{1, 2, 3\}$ and $d, e, f$ are in $\{4, 5, 6\}$. The WZW term in Eq. (V.20) calculates the linking number between the two charge operators,

$$\frac{2\pi \mathbf{i}}{\Omega_5} \int_Y \epsilon_{abcdef} n^a dn^b \wedge dn^c \wedge dn^d \wedge dn^e \wedge dn^f = 2\pi \mathbf{i} \int_Y \omega_\mathsf{U}^{(2)} \wedge d\omega_\mathsf{SO}^{(2)} = 2\pi \mathbf{i} \mathsf{Lk}(\mathbb{S}_\mathsf{U}^2, \mathbb{S}_\mathsf{SO}^2). \qquad \text{(V.21)}$$

If fixing the position of one charge operator $\omega_\mathsf{U}^{(2)}$, and move the other charge operator $\omega_\mathsf{SO}^{(2)}$ around the fixed position one $\omega_\mathsf{U}^{(2)}$, then the WZW term describes that the worldsheet of the moving monopole $\omega_\mathsf{SO}^{(2)}$ detects the flux of $\omega_\mathsf{U}^{(2)}$. And the WZW term assigns phase to their linking number, this demonstrates essentially the intertwinement of charge operators of topological defects.

Since the nonlinear sigma model with the WZW term can be viewed as the boundary theory of the bulk SPT, one can instead see the intertwinement in the bulk SPT. Once coupling the charge operators of the defects to 2-form background gauge fields $B^{(2)}, C^{(2)}$, the bulk SPT is described by, $\frac{1}{\pi} \int_Y B^{(2)} \wedge dC^{(2)}$. The linking between the surface operators $e^{\mathbf{i} \oint B}, e^{\mathbf{i} \oint C}$ is also given by the above linking number [83].

These bosonic fields can be further embedded into $G/K$ by embedding the generators $T^A, \tilde{T}^A$ into the generators of $G/K$. In the following section Sec. VI C, we will couple these Goldstone boson fields to the fermionic field and construct the fermionic sigma model. The fermionic sigma model also shows the intertwinement of the charge operators is related to the higher linking number of two $\mathbb{S}^2$s in $\mathbb{S}^5$.

## VI. FERMIONIC SIGMA MODEL AND WZW TERM

In this section, we construct the fermion model that reproduces NLSM with WZW for general homogeneous space $G/H$. The fermions are coupled to the fluctuating bosonic fields living in $G/H$. After integrating out the fermion fields [29, 30], the resulting effective action is the NLSM with WZW given in Sec. V. We call such models as fermionic sigma model and they provide an alternative insight into the intertwinement of symmetry defects in higher dimensions.

### A.   General $G$-symmetric action and fermionic sigma model

The bosonic fields introduced in Sec. II B transforms nonlinearly under the global symmetry $G$ as in Eq. (II.6). We can consider a field $\chi$ that transforms under $G$ as,

$$\chi \xrightarrow{g} \chi' = D(h^{-1}(g,\pi))\chi, \tag{VI.1}$$

with some representation $D$. The $\chi$ field is like a representative point on the coset, and it can be rotated to the general one by the Goldstone boson field. The $\chi$ field can be converted into the field $\psi = U(\pi)\chi$ that transforms as an ordinary linear transformation under $G$,

$$\psi(x) \xrightarrow{g} \psi'(x) = D(U(\pi'))\chi'(x)$$
$$= D(g^{-1}U(\pi)h(g,\pi))D(h^{-1}(g,\pi))\chi(x) = D(g^{-1})D(U(\pi))\chi(x) = D(g^{-1})\psi(x) \tag{VI.2}$$

We can also introduce the covariant derivative, $\mathcal{D}_V\chi$. Under the $G$ transformation it becomes,

$$\mathcal{D}_V\chi = (\partial_\mu + V)\chi \xrightarrow{g} \partial_\mu(h^{-1}(\pi,g)\chi) + (V + h^{-1}dh)h^{-1}(\pi,g)\chi = h^{-1}(\pi,g)\mathcal{D}_V\chi. \tag{VI.3}$$

Meanwhile, as in Eq. (II.13), $\phi \xrightarrow{g} h^{-1}\phi h$. Therefore, the general $G$ invariant action can be constructed by $\chi, \mathcal{D}_V\chi, \phi$, which is also invariant under the unbroken symmetry $H$ [18–20].

To construct the fermionic sigma model, we introduce a mass matrix $M_0$ as a reference point, and it satisfies,

$$hM_0h^{-1} = M_0. \tag{VI.4}$$

For example, $M_0 = \text{diag}(1,..,1,-1,...,-1)$ with $n$ times $+1$, $m$ times -1, is the matrix that breaks $\mathsf{SO}(n+m) \to \mathsf{SO}(n) \times \mathsf{SO}(m)$. The $\bar{\chi}M_0\chi$ is then $G$-symmetric,

$$g: \bar{\chi}M_0\chi \to \bar{\chi}D(h^{-1})^{-1}M_0D(h^{-1})\chi = \bar{\chi}M_0\chi \tag{VI.5}$$

Upon rewriting the term in the $\psi$ basis, the $G$-symmetric action is,

$$\bar{\psi}\mathbf{i}\gamma^\mu\partial_\mu\psi + \bar{\psi}U(\pi)M_0U(\pi)^{-1}\psi. \tag{VI.6}$$

where $\psi$ is the complex fermion that transforms linearly under $G$. Eq. (VI.6) is the general fermionic sigma model where the fermion mass manifold $G/H$ is parameterized by the bosonic field $U$.

## B. Reproducing the WZW term from the fermionic sigma model

We follow the Ref. [29] and recent presentation in Ref. [30] to derive the Wess-Zumino-Witten term by integrating out the fermion, the partition function of the anomalous theory depends on the Goldstone boson field is,

$$\mathcal{Z}_{\mathcal{T}}[U] = \int \mathcal{D}\bar{\psi}\mathcal{D}\psi e^{-\mathcal{S}[\bar{\psi},\psi,U]}, \tag{VI.7}$$

$$\mathcal{S}[\bar{\psi},\psi,U] = \int d^n x \ \bar{\psi}(\mathbf{i}\gamma^\mu\partial_\mu + \mathbf{i}mU(\pi)M_0 U(\pi)^\dagger)\psi$$

$$\equiv \int d^n x \ \bar{\psi}(\mathbf{i}\slashed{\partial} + \mathbf{i}mM^U)\psi \equiv \int d^n x \ \bar{\psi}\hat{\mathcal{D}}\psi, \tag{VI.8}$$

where $M^U \equiv U(\pi)M_0 U(\pi)^{-1}$ and $\mathcal{Z}_{\mathcal{T}}[U]$ contains the kinetic term and possible Wess-Zumino-Witten term of the Goldstone boson. Following the standard derivation, the WZW action is,

$$S_{\text{WZW}} = -\frac{1}{2\pi}\frac{1}{(4\pi)^{d/2}}\frac{\Gamma(\frac{d}{2}+1)}{\Gamma(d+1)}\int_Y \mathrm{d}u d^n x \ \mathrm{tr}\left(\prod_{i=1}^{n}(\gamma^{\mu_a}\partial_{\mu_a}M^U)M^{U\dagger}\partial_u M^U\right) \tag{VI.9}$$

where $u$ is the extra coordinate on $Y$, $\partial Y = X$, $\Gamma(z)$ is the Gamma function $\Gamma(n+1) = n!$ for integer $n$. Since the WZW term is written locally in 1 higher dimension than the spacetime manifold, we have extended $M^U$ as the map from $Y$ to the mass manifold. After straightforward calculation, the WZW term for $G/H$ is in general given by,

$$\Gamma^{(d+1)}(U) = -\frac{1}{2\pi}\frac{2^{\lfloor d/2\rfloor}}{(4\pi)^{d/2}}\frac{\Gamma(\frac{d}{2}+1)}{\Gamma(d+2)}\mathrm{tr}\left([M_0, U^{-1}dU]^{d+1}M_0^\dagger\right) \tag{VI.10}$$

where $\lfloor x\rfloor$ is the floor function. Recalling that $U^{-1}dU$ can be decomposed into $\mathfrak{h}$ and $\mathfrak{f}$ parts, $U^{-1}dU = V + \phi$, and $[M_0, T^\alpha] = 0, T^\alpha \in \mathfrak{h}$, we have,

$$[M_0, U^{-1}dU] = [M_0, \phi] = [M_0, \theta^a T^a], a \in \mathfrak{f} \tag{VI.11}$$

It turns out, for example,

$$\Gamma^{(5)} = -\frac{1}{480\pi^3}\mathrm{tr}\left([M_0, U^{-1}dU]^5 M_0^\dagger\right) \quad \in H^5(G/H, \mathbb{R}) \tag{VI.12}$$

$$= -\frac{\mathbf{i}}{16\pi^3}\mathrm{tr}\left(\phi W^2 + \frac{1}{2}\phi\{W, \mathcal{D}_V\phi\} + \frac{2}{3}W\phi^3 + \frac{1}{3}\phi(\mathcal{D}_V\phi)^2 + \frac{1}{2}\phi^3\mathcal{D}_V\phi + \frac{1}{5}\phi^5\right) \tag{VI.13}$$

This shows that the fermionic sigma model in Eq. (VI.6) reproduces the WZW term for $G/H$ homogeneous space.

## C. Fermionic sigma model and intertwinement of mass manifolds

In this section, we present the construction of a fermionic sigma model that could reproduce the WZW term in Sec. V B. There are two types of topological defects in the symmetry breaking phases,

both of them are characterized by the charge operators as the generators of the second cohomology $H^2(G/K)$. We then consider embedding two $\mathbb{S}^2$s into $G/K$, the linking number of these two spheres is the degree of the mapping from two $\mathbb{S}^2$ to $\mathbb{S}^5$. More explicitly, to illustrate the intertwinement of the topological defects, we consider the mapping in Eq. (V.16), $\mathbb{S}^2_{\mathsf{U}} \sqcup \mathbb{S}^2_{\mathsf{SO}} \xrightarrow{g} \mathbb{S}^5 \xrightarrow{h} \widehat{G/K}$.

We are focusing on the case where the global symmetry is $\mathsf{SO}(8)$, the generalization of this construction to $\mathsf{SO}(2n)$ can be obtained by embedding $\mathsf{SO}(8) \hookrightarrow \mathsf{SO}(2n)$. The embedding of two disjoint $\mathbb{S}^2$s into $G/K$ is obtained by considering two commuting $\mathfrak{so}(3)$s and modulo the $\mathfrak{so}(2)$ subalgebra.

The Goldstone boson fields in Eq. (V.18) can be used to rotate the mass matrix and coupled to the fermions. Therefore, we can construct the fermionic sigma model that reproduces the WZW term, or the charge operators of the topological defects. Here we consider the fermions that are transformed under vector representation of the global flavor symmetry $\mathsf{SO}(2n)$ and the mass matrix can be an antisymmetric or symmetric representation of $\mathsf{SO}(2n)$.

As noted in Sec. V B and [25, 26], the higgs fields $\Phi_{\mathbf{45}}, \Phi_{\mathbf{54}}$ which are used to approach GG and PS phase have different symmetry properties, they are symmetric and antisymmetric respectively. The mass matrix of the fermion model can be chosen in a way that aligns with the symmetry properties, once integrating out the fermion fields, the corresponding charge operators of the topological defects could match with the symmetry constraints of the higgs fields $\Phi_{\mathbf{45}}, \Phi_{\mathbf{54}}$. We are considering this symmetry constraint also applies to the $\mathsf{SO}(2n), n \geq 4$ model.

However, the $\mathfrak{so}(3)$ matrices considering in Eq. (V.17) are all antisymmetric. The way to render the antisymmetric matrix to a symmetric one is to Kronecker product additional $\sigma^2$ to the antisymmetric matrices, to preserve the antisymmetry, one needs to Kronecker product additional $\sigma^0$ to the antisymmetric matrices. Due to the symmetry constraint [25, 26], we would like to construct one set with all symmetric matrices and the other set with all antisymmetric matrices.

$$Sym : \{\sigma^{220}, \sigma^{212}, \sigma^{232}\}, \quad Asym : \{\sigma^{002}, \sigma^{021}, \sigma^{023}\}. \tag{VI.14}$$

Hence, the first set of $\mathsf{SU}(2)$ matrices is symmetric, the second set is antisymmetric. The above matrices are ready to couple to complex fermions. In the Majorana basis, the mass matrix should be antisymmetric, and the general form of the mass matrices in $4d$ is,

$$M = \sigma^{21} \otimes S_1 + \sigma^{23} \otimes S_2, \tag{VI.15}$$

where $S_i$ are the symmetric matrices, $\sigma^{21}, \sigma^{23}$ are the $\gamma$ matrices. We need further add indices to

the two sets of matrices and make them symmetric,

$$Sym : \{\sigma^{0220}, \sigma^{0212}, \sigma^{0232}\}, \quad Sym : \{\sigma^{2002}, \sigma^{2021}, \sigma^{2023}\}. \tag{VI.16}$$

It is convenient to block diagonalize the matrices by doing the unitary transformation $e^{\mathbf{i}\frac{\pi}{4}\sigma^{1200}}$,

$$M \to e^{\mathbf{i}\frac{\pi}{4}\sigma^{1332}} M e^{-\mathbf{i}\frac{\pi}{4}\sigma^{1332}} : Sym : \{\sigma^{0220}, \sigma^{0212}, \sigma^{0232}\}, \quad Sym : \{\sigma^{3202}, \sigma^{3221}, \sigma^{3223}\}. \tag{VI.17}$$

One can freely choose the representative matrix in each set, and the other matrices can be obtained by doing the $\mathsf{SU}(2)$ transformation,

$$M_0^{\mathsf{SO}} = \sigma^{0220}, T^a = \{\sigma^{0012}, \sigma^{0032}\}, \tag{VI.18}$$

$$M_0^{\mathsf{U}} = \sigma^{3202}, \tilde{T}^a = \{\sigma^{0021}, \sigma^{0023}\}. \tag{VI.19}$$

Since the mass matrices are block-diagonal, one can rotate the upper block or the lower block separately by

$$M_0^{\mathsf{SO}} = \sigma^{0220}, T_\pm^a = \{\frac{\sigma^{0012} \pm \sigma^{3012}}{2}, \frac{\sigma^{0032} \pm \sigma^{3032}}{2}\} \tag{VI.20}$$

$$M_0^{\mathsf{U}} = \sigma^{3202}, \tilde{T}_\pm^a = \{\frac{\sigma^{0021} \pm \sigma^{3021}}{2}, \frac{\sigma^{0023} \pm \sigma^{3023}}{2}\}. \tag{VI.21}$$

Hence, we obtain the map from $S^2$ to the $\mathsf{SU}(2)$ mass matrices,

$$(\theta_1, \phi_1) \in S^2 \to M_\pm^{\mathsf{SO}} = U_1 M_0^{\mathsf{SO}} U_1^{-1} \in \mathsf{SU}(2), \quad U_1 = e^{\mathbf{i}(\theta_1 \sin\phi_1, \theta_1 \cos\phi_1) \cdot (T_\pm^1, T_\pm^2)^{\mathsf{T}}} \tag{VI.22}$$

$$(\theta_2, \phi_2) \in S^2 \to M_\pm^{\mathsf{U}} = U_2 M_0^{\mathsf{U}} U_2^{-1} \in \mathsf{SU}(2), \quad U_2 = e^{\mathbf{i}(\theta_2 \sin\phi_2, \theta_2 \cos\phi_2) \cdot (\tilde{T}_\pm^1, \tilde{T}_\pm^2)^{\mathsf{T}}} \tag{VI.23}$$

where $M_\pm^{\mathsf{SO}}, M_\pm^{\mathsf{U}}$ satisfy $[M_\pm^{\mathsf{SO}}, M_\pm^{\mathsf{U}}] = 0$. We first find that the charge operators of the topological defects can be reproduced by the fermions coupled to the mass manifolds and evaluated on a submanifold,

$$M_\pm^{\mathsf{SO}} dM_\pm^{\mathsf{SO}} dM_\pm^{\mathsf{SO}} = vol_{S^2} \sigma_\pm^{200}, \quad M_\pm^{\mathsf{U}} dM_\pm^{\mathsf{U}} dM_\pm^{\mathsf{U}} = vol_{S^2} \sigma_\pm^{200} \tag{VI.24}$$

where $vol_{S^2} = \sin\theta_i d\theta_i d\phi_i$ is the volume form of the $S^2$. More interestingly, if we interpolate the mass matrix in Eq. (VI.15) by,

$$\mathsf{SU} \ni M(u, \theta_1, \phi_1, \theta_2, \phi_2) = \sigma^{21} \otimes u M^{\mathsf{SO}} + \sigma^{23} \otimes \sqrt{1-u^2} M^{\mathsf{U}}, \tag{VI.25}$$

such that $M(0) = \sigma^{23} \otimes M^{\mathsf{U}}, M(1) = \sigma^{21} \otimes M^{\mathsf{SO}}$. Note that the two mass matrices play different roles, one is the identity mass, and the other one relates to the chiral mass. When integrating out the fermion fields, the WZW term is,

$$\begin{aligned} S_{\mathrm{WZW}} &= \frac{2\pi}{960\pi^3} \int_{S^2 \times S^2 \times I} \mathrm{tr}(M^{-1}dM)^5 \\ &= \frac{2\pi}{960\pi^3} \int_{S^2 \times S^2 \times I} 120\mathrm{tr}(\sigma^{000}) u^2 \sqrt{1-u^2} \sin\theta_1 \sin\theta_2 d\theta_1 d\phi_1 d\theta_1 d\phi_1 du. \end{aligned} \tag{VI.26}$$

When evaluating the WZW term on an interval $u \in [0,1]$,

$$S_{\text{WZW}} = \frac{2\pi}{960\pi^3} \int_0^1 \int_{S^2} \int_{S^2} 120 \text{tr}(\sigma^{000}) u^2 \sqrt{1-u^2} \sin\theta_1 \sin\theta_2 d\theta_1 d\phi_1 d\theta_1 d\phi_1 du \qquad \text{(VI.27)}$$

$$= \int_{S^2 \times S^2} \frac{-\mathbf{i} \sin\theta_1 \sin\theta_2}{8\pi} d\theta_1 d\phi_1 d\theta_2 d\phi_2 \qquad \text{(VI.28)}$$

$$= 2\pi\mathbf{i} = 2\pi\mathbf{i}\mathsf{Lk}(\mathbb{S}^2, \mathbb{S}^2), \qquad \text{(VI.29)}$$

where $\mathsf{Lk}(\mathbb{S}^2, \mathbb{S}^2)$ is the linking number of two $\mathbb{S}^2$ in $\mathbb{S}^5$ [78, 83]. This WZW term corresponds to $\phi W_1 W_2$ in the previous section, where $W_1, W_2$ are the curvature of the two 2-spheres corresponds to the generator of $H^2(G/H_i, \mathbb{R})$ and $\phi$ is $\mathfrak{f}$-valued 1-form, but interestingly, $\phi$ relates to the chiral rotation $\mathsf{U}(1)$ in the global symmetry of the fermionic sigma model, which corresponds to the exchange of two symmetry defects in the $G/K$ NLSM in Eq. (V.14).

## VII.   SUMMARY AND COMMENTS

*a.  Summary*   We propose the nonlinear sigma model with target space $G/K$ and Wess-Zumino-Witten term as the general description of deconfined quantum critical point theory, based on the very important features of the symmetry defects and their intertwinement in the DQCP theories. We show the topological defects in $G/K$ precisely correspond to the symmetry defects in each spontaneous symmetry breaking phase in the DQCP phase diagram. The WZW term decorates the symmetry defects in one SSB phase with the charge of the broken symmetry of the other SSB phase. By proliferating the symmetry defects, the broken symmetry of one SSB phase is restored but the additional charge breaks the symmetry, leading to the other SSB phase.

We connect this NLSM description with the ordinary 't Hooft anomaly matching argument by explicitly calculating the gauged WZW term and its corresponding bulk SPT. We find the bulk SPT is in general described by relative Chern-Simons term when the anomalous UV symmetry $G$ is spontaneously broken to non-zero subgroup $H$ (which can be anomalous or non-anomalous). We provide an alternative fermionic sigma model that reproduces the NLSM with the WZW term. This alternative fermionic model gives insight into the detailed global symmetry actions.

We apply our framework to several examples - first revisit the ordinary 2+1d DQCP between Néel and VBS phases. Then motivated by recent works on deconfined quantum criticality among different grand unified theories [25, 26], we studied the deconfined quantum critical theories between two SSB phases with unbroken symmetries $H_{\mathsf{SO}} = \mathsf{SO}(2m) \times \mathsf{SO}(2n-2m)$ and $H_{\mathsf{U}} = \mathsf{U}(n)$, and they come from the theory with $G = \mathsf{SO}(2n)$ global symmetry by condensing the order parameters.

Applying the $G/K$ NLSM description ($K = \mathsf{U}(m) \times \mathsf{U}(n-m)$), we are able to find operators that correspond to the symmetry defects in both SSB phases, due to $\pi_2(\frac{G}{\mathsf{U}(m)\times\mathsf{U}(n-m)}) = \mathbb{Z} \oplus \mathbb{Z}$. It is interesting because the symmetry defect in the SSB phase with unbroken symmetry $H_{\mathsf{SO}}$ is Grassmannian manifold and has $\mathbb{Z}_2$ valued topological charge, which does not have a corresponding de Rham cohomology description. Embedding $G/H_{\mathsf{SO}}$ into larger space $G/K$ is reminiscent of finding non-perturbative $\mathsf{SU}(2)$ anomaly by embedding $\mathsf{SU}(2) \hookrightarrow \mathsf{SU}(3)$, and the non-perturbative anomaly associated with $\pi_4(\mathsf{SU}(2)) = \mathbb{Z}_2$ can be seen from $\mathsf{SU}(3)$ WZW term via perturbative calculation. Then we construct the WZW term and examine the corresponding anomaly which descends from the $\mathsf{SO}(2n)$ anomaly [25, 26, 28].

Furthermore, the symmetry defects in this complicated homogeneous space can be understood by examining the embedding $\mathbb{S}^2 \times \mathbb{S}^2 \xrightarrow{f} G/K$. Hence, the $G/K$ NLSM becomes the ordinary $\mathsf{O}(6)$ nonlinear sigma model with the WZW term. The first and last three components of the $\mathsf{O}(6)$ vector describe the 2-spheres corresponding to different symmetry defects. The WZW term then assigns the phase to the linking of the two 2-spheres in $\mathbb{S}^5$.

We provide an alternative fermionic sigma model to reproduce the NLSM. The fermions are coupled to the fluctuating bosonic fields living in the homogeneous space $G/K$, when integrating out the fermions, the resulting effective action is the $G/K$ NLSM with level-1 WZW term. As an example, we embed the two 3-component unit vectors into $G/K$ and construct the fermionic model of $\mathsf{O}(6)$ NLSM. We should point out that since the $\mathsf{SO}(6)$ is explicitly broken down to $\mathsf{SO}(3)\times\mathsf{SO}(3)$, the chiral $\mathsf{U}(1)$ rotation in the fermion model is crucial to get the correct linking between symmetry defects in different SSB phases. This rotation is in $\mathsf{SO}(6)$ but not in $\mathsf{SO}(3) \times \mathsf{SO}(3)$.

*b. Comments* The $G/K$ NLSM with WZW description discussed in this paper is applicable to any dimensions and different continuous symmetries of DQCP theory. However, this description focuses on the kinematics of the DQCP theory, namely the symmetry defects, their intertwinement, and 't Hooft anomaly. The dynamics of the DQCP theory is much more complicated - the operator contents and their scaling dimensions are not universal, and the renormalization group schemes vary from different dimensions and different models. Nevertheless, the symmetry of the $G/K$ NLSM would imply infrared duality of gauge theories, for example, the discrete symmetry that exchanges two types of symmetry defects becomes particle-vortex like duality of gauge theories [12, 59, 84]. Furthermore, the duality between different quantum field theories relates the operator contents and set the constraints on the scaling dimensions which reveals information on dynamics [12, 85, 86].

Despite the difficulty in extracting specific dynamical information, our proposal captures the

essential features for which the DQCP is beyond ordinary symmetry-breaking transition. In this point of view, the DQCP is not rare, and it can be more ubiquitous if incorporating higher-form symmetry [87–91], categorical symmetry [92, 93], and loop group symmetry for the system with a fermi surface [94–96]. The ongoing exploration of non-invertible symmetries should also have their corresponding DQCP theory provided the symmetries have mixed anomaly [97–99]. One can also apply the current approach to understand multicritical point joined by several SSB phases.

**Acknowledgements** I'm deeply grateful to Yi-Zhuang You for numerous discussions through the whole project and for his generosity to introduce the example of DQCP in GUT to me. I am also grateful to John McGreevy for his series of lectures on quantum matters. I would like to thank Yi-Zhuang You, John McGreevy, Tarun Grover, Juven Wang and Zhengdi Sun for various discussions and conversations related to this topic.

## Appendix A: de Rham cohomology of Lie groups and homogeneous spaces

### 1.   de Rham complex

Let $e_i$ denote the basis for the Lie algebra $\mathfrak{g}$ and $\theta^i$ for the 1-forms for $\mathfrak{g}^*$, which is the dual space of $\mathfrak{g}$. The $p$-forms on $\mathfrak{g}$ are the alternating multi-linear maps $\omega : \mathfrak{g} \times ... \times \mathfrak{g} \to \mathbb{R}$. For $x_a$ being the basis for space $V$, the $V$-valued $p$-form on $\mathfrak{g}$, $\omega \in \Lambda^p(\mathfrak{g}^*, V)$ can be written as,

$$\omega = A^\alpha_{i_1,..,i_p} x_\alpha \theta^{i_1} \wedge ... \wedge \theta^{i_p}. \tag{A.1}$$

For example, the Maurer-Cartan 1-forms are Lie algebra valued 1-forms,

$$\theta = \theta^A T^A \in \Lambda^1(\mathfrak{g}^*, \mathfrak{g}). \tag{A.2}$$

The exterior derivative sends the $p$-forms to $p+1$-forms $\mathrm{d} : \Lambda^p(\mathfrak{g}^*, V) \to \Lambda^{p+1}(\mathfrak{g}^*, V)$ and follows the rule,

$$\mathrm{d}\theta^i = -\frac{1}{2} f^i_{jk} \theta^j \wedge \theta^k, \quad \mathrm{d}x_\alpha = B^\beta_{\alpha i} x_\beta \theta^i, \tag{A.3}$$

where $f^i_{jk}$ is the structure factor for the Lie algebra and $B^\beta_{\alpha i}$ is a certain linear map for the $V$-space. For $\mathfrak{h} \subset \mathfrak{g}$ being a subalgebra of $\mathfrak{g}$, the relative cochain is given by,

$$\Lambda^p(\mathfrak{g}^*, \mathfrak{h}, V) = \{\omega \in \Lambda^p(\mathfrak{g}^*, V) | i_y(\omega) = 0 \text{ and } i_y(\mathrm{d}\omega) = 0, \forall y \in \mathfrak{h}\}. \tag{A.4}$$

where $i_y$ is the interior product, in other words, the forms $\omega$s as well as $\mathrm{d}\omega$s do not contain $\theta^i$s from the subalgebra $\mathfrak{h}$ parts, and the forms are invariant under adjoint action of $H$. In the following, we

will mainly consider $\Lambda^p(\mathfrak{g}^*, \mathbb{R})$ and $\Lambda^p(\mathfrak{g}^*, \mathfrak{h}, \mathbb{R})$ which is relevant to the Wess-Zumino-Witten term for $G, G/H$ and other topological terms in the physical actions, thus, no $x_\alpha$ dependence.

The condition to construct relative cochain implies

$$\mathcal{L}_y \omega = (d i_y + i_y d)\omega = 0 \tag{A.5}$$

where $\mathcal{L}_y$ is the Lie derivative with respect to $y$, the relative cochain is then given by,

$$\Lambda^p(\mathfrak{g}^*, \mathfrak{h}, \mathbb{R}) = \{\omega \in \bigwedge^p (\mathfrak{g}/\mathfrak{h})^* | \mathcal{L}_y \omega = 0, \forall y \in \mathfrak{h}\}. \tag{A.6}$$

the Lie derivative action is explicitly expressed in terms of the components of the Maurer-Cartan 1-form,

$$\mathcal{L}_y \omega^{(n)} = -\sum_{i=1}^{n} \frac{1}{n!} \omega_{a_1, \ldots, a_n} f^{b_j, y, a_j} \theta^{a_1} \wedge \ldots \wedge \theta^{b_j} \wedge \ldots \wedge \theta^{a_n} = 0. \tag{A.7}$$

The relative cochain can be constructed by first finding the space spanned by $\bigwedge^p (\mathfrak{g}/\mathfrak{h})^*$ and then using $\mathcal{L}_y, y \in H$ iteratively to eliminate non-invariant bases.

## 2. Cohomology

A $p$-form $\omega \in \Lambda^p(\mathfrak{g}^*, \mathfrak{h}, \mathbb{R})$ is **closed**, if $d\omega = 0$; and **exact** if it can be expressed by a $(p-1)$-form $\eta$ by $\omega = d\eta$. Since $d^2 = 0$ for any differential forms $\omega$, the exact forms are necessarily closed but the closed forms can be non-exact.

The cohomology $H^*(G, \mathbb{R})$ and $H^*(G/H, \mathbb{R})$ measures the closed forms not being exact. Consequently, the $p$-forms cannot be expressed locally in $p - 1$ dimension by Stokes theorem.

We explicitly calculate the cohomology group using the basis of the general $p$-form generated by the exterior product of the 1-form components $\theta^i$s. For example, the basis for the 2-forms in $\Lambda^2(\mathfrak{g}^*, \mathbb{R})$ are,

$$\{\theta^1 \wedge \theta^2, \theta^1 \wedge \theta^3, \ldots, \theta^{\dim(G)-1} \wedge \theta^{\dim(G)}\}. \tag{A.8}$$

The exterior derivative can therefore be expressed as a matrix $\tilde{d}_{ab}$,

$$d\{(\theta^i)^{\wedge p}\}_a = \tilde{d}^p_{ab}\{(\theta^i)^{\wedge (p+1)}\}_b \tag{A.9}$$

The matrix $\tilde{d}^p_{ab}$ is a $\binom{\dim(G)}{p} \times \binom{\dim(G)}{p+1}$ matrix. The subspace of the closed $p$-forms $C^p$ is the null-space or the kernel of the matrix $(\tilde{d}^p_{ab})^T$,

$$\text{subspace of the closed } p\text{-forms: } C^p = \ker (\tilde{d}^p_{ab})^\mathsf{T} \tag{A.10}$$

The subspace of the exact $p$-forms $Z^p$ is the the orthogonal complement of the kernel of $(\tilde{d}_{ab}^{p-1})$,

$$\text{subspace of the exact } p\text{-forms: } Z^p = (\ker \; \tilde{d}_{ab}^{p-1})^\perp \tag{A.11}$$

This can be obtained by Gaussian elimination of the matrix $\tilde{d}_{ab}^{p-1}$. Therefore, the cohomology is the quotient,

$$H^p = \frac{\ker \; (\tilde{d}_{ab}^{p})^\intercal}{(\ker \; \tilde{d}_{ab}^{p-1})^\perp} \tag{A.12}$$

Algorithmically, we denoted the space of closed $p$-form as $C^p$ and exact $p$-form as $Z^p$, they are both matrices, and the cohomology is,

$$[\ker \; C^p \cdot (Z^p)^\intercal \cdot Z^p \cdot (C^p)^\intercal] \cdot C^p \tag{A.13}$$

For the relative cochain, one needs to further impose the constraint in Eq. (A.6). This constraint corresponds to dropping the basis which contains indices corresponding to that in $\mathfrak{h}$ and invariant under the adjoint transformation of $H$. One can start with the basis constructed by wedge product of $\theta^a$s, where $a \in \mathfrak{g}/\mathfrak{h}$, and then use the Lie derivative for each $h \in H$ to eliminate non-zero bases.

### 3. Examples

Using the de Rham cohomology, we are able to calculate the following examples. And we compare our results with the general results which are cited from [37] if not citing others.

**SU(4):** Our calculation gives,

$$H^3(\mathsf{SU}(4), \mathbb{R}) = \mathbb{R}, \quad H^5(\mathsf{SU}(4), \mathbb{R}) = \mathbb{R}. \tag{A.14}$$

In general, $H^*(\mathsf{SU}(n)) = \Lambda(e_3, e_5, ..., e_{2n-1})$, where $e_i \in H^i(\mathsf{SU}(n), \mathbb{R})$, the cohomology ring is generated by wedge product.

**SO(6)/(SO(4)$\times$ SO(2))**, $\dim = 15 - 7 = 8$ even dimensional: Our calculation gives,

$$H^2 \left( \frac{\mathsf{SO}(6)}{\mathsf{SO}(4) \times \mathsf{SO}(2)}, \mathbb{R} \right) = \mathbb{R}, \quad H^4 \left( \frac{\mathsf{SO}(6)}{\mathsf{SO}(4) \times \mathsf{SO}(2)}, \mathbb{R} \right) = \mathbb{R} \oplus \mathbb{R}. \tag{A.15}$$

In general,

$$H^*(\mathsf{SO}(2n+2)/(\mathsf{SO}(2n) \times \mathsf{SO}(2))) = (1 + t^{2n})(1 + t^2 + t^4 + ... + t^{2n}) \tag{A.16}$$

For $n = 2$, $H^*(\mathsf{SO}(6)/(\mathsf{SO}(4) \times \mathsf{SO}(2))) = 1 + t^2 + 2t^4 + t^6 + t^8$, where $t^n$ corresponds to the degree $n$ generator, $2t^4$ means 2 independent degree 4 generators. In general, the Poincare polynomial for

$H^*(\frac{\mathsf{SO}(2n)}{\mathsf{SO}(2k)\times\mathsf{SO}(2n-2k)})$ is in [100],

$$
\begin{array}{c|c}
H^*(\frac{\mathsf{SO}(8)}{\mathsf{SO}(4)\times\mathsf{SO}(4)}) & 1+3t^4+4t^8+... \\
\hline
H^*(\frac{\mathsf{SO}(10)}{\mathsf{SO}(4)\times\mathsf{SO}(6)}) & 1+2t^4+t^6+3t^8...
\end{array}
\tag{A.17}
$$

also,

$$
\begin{array}{c|c}
H^*(\frac{\mathsf{SO}(8)}{\mathsf{SO}(2)^4}) & 1+4t^2+9t^4+... \\
\hline
H^*(\frac{\mathsf{SO}(10)}{\mathsf{SO}(2)^5}) & 1+5t^2+14t^4...
\end{array}
\tag{A.18}
$$

Our explicit calculation of cohomology up to 4 degree agrees with the general results.

**SO(6)/U(3)**: Our calculation gives,

$$
H^2\left(\frac{\mathsf{SO}(6)}{\mathsf{U}(3)},\mathbb{R}\right)=\mathbb{R},\quad H^4\left(\frac{\mathsf{SO}(6)}{\mathsf{U}(3)},\mathbb{R}\right)=\mathbb{R}.
\tag{A.19}
$$

In general, $H^*(\mathsf{SO}(2n)/\mathsf{U}(n))=\Delta(e_2,e_4,...,e_{2n-2})$.

Our calculation of other cohomology of cosets with $G=\mathsf{SO}$,

$$
\begin{array}{c|c|c|c|c}
 & \frac{\mathsf{SO}(8)}{\mathsf{SO}(4)\times\mathsf{SO}(4)} & \mathsf{SO}(8)/\mathsf{U}(4) & \frac{\mathsf{SO}(10)}{\mathsf{SO}(4)\times\mathsf{SO}(6)} & \mathsf{SO}(10)/\mathsf{U}(5) \\
\hline
H^2(G/H,\mathbb{R}) & \varnothing(\mathbb{Z}_2) & \mathbb{R} & \varnothing(\mathbb{Z}_2) & \mathbb{R}
\end{array}
\tag{A.20}
$$

The torsion $\mathbb{Z}_2$ of $H^2(\frac{\mathsf{SO}(8)}{\mathsf{SO}(4)\times\mathsf{SO}(4)})$ cannot be detected by de Rham cohomology.

*a. Cohomology of $G/K$* The cohomology of $G/K$ is relevant to the symmetry defects in spontaneously symmetry-breaking phases. The Lie group $K$ is generated by the Lie algebra $\mathfrak{k}=\mathfrak{h}_1\cap\mathfrak{h}_2$, and our cohomology calculation gives,

$$
\begin{array}{c|c|c|c}
 & \frac{\mathsf{SO}(6)}{\mathsf{SO}\mathsf{SO}\cap\mathsf{U}} & \frac{\mathsf{SO}(8)}{\mathsf{SO}\mathsf{SO}\cap\mathsf{U}} & \frac{\mathsf{SO}(10)}{\mathsf{SO}\mathsf{SO}\cap\mathsf{U}} \\
\hline
H^2(G/H,\mathbb{R}) & \mathbb{R}\oplus\mathbb{R} & \mathbb{R}\oplus\mathbb{R} & \mathbb{R}\oplus\mathbb{R}
\end{array}
\tag{A.21}
$$

where $\mathbb{R}\oplus\mathbb{R}$ in $H^2(\mathsf{SO}(6)/(\mathsf{SO}\mathsf{SO}\cap\mathsf{U}))$ are the same generators of $H^2(G/H_1),H^2(G/H_2)$. For $\mathsf{SO}(8),\mathsf{SO}(10)$, one is the same generator of $\mathsf{SO}/\mathsf{U}$, another is the new from both $\mathsf{SO}/\mathsf{U}$ and $\mathsf{SO}/\mathsf{SO}$ parts.

*b. Other cosets* **SU(4)/SO(4)**: Our calculation shows,

$$
H^4\left(\frac{SU(4)}{SO(4)},\mathbb{R}\right)=\mathbb{R},\quad H^5\left(\frac{SU(4)}{SO(4)},\mathbb{R}\right)=\mathbb{R}
\tag{A.22}
$$

In general,

$$
H^*\left(\frac{SU(n)}{SO(n)},\mathbb{R}\right)=\begin{cases}\Lambda(e_5,...,e_{4m+1}) & n=2m+1 \\ \Lambda(e_5,...,e_{4m-2})\otimes\Delta(e_{2m}) & n=2m\end{cases}
\tag{A.23}
$$

**SO(6)/(SO(3)$\times$ SO(3))**, $\dim=15-6=9$ odd dimensional: Our calculation shows,

$$
H^4\left(\frac{SO(6)}{SO(3)\times SO(3)},\mathbb{R}\right)=\mathbb{R},\quad H^5\left(\frac{SO(6)}{SO(3)\times SO(3)},\mathbb{R}\right)=\mathbb{R}
\tag{A.24}
$$

## Appendix B: Cartan homotopy formula

### 1. Review of Cartan homotopy method

If two connections are of the same bundle, one can consider the interpolation [58],

$$\mathcal{A}_t = \mathcal{A}_0 + t(\mathcal{A}_1 - \mathcal{A}_0), \quad \mathcal{F}_t \equiv d\mathcal{A}_t + \mathcal{A}_t^2 \tag{B.1}$$

Another useful formula,

$$\mathcal{D}_\mathcal{A}\eta = d\eta + [\mathcal{A}, \eta], \tag{B.2}$$

$$[\eta^{(p)}, \omega^{(q)}] = \eta^{(p)} \wedge \omega^{(q)} - (-1)^{pq}\omega^{(q)} \wedge \eta^{(p)} \tag{B.3}$$

Define the anti-deriviative operator $\ell_t$,

$$\ell_t\mathcal{A}_t = 0, \quad \ell_t\mathcal{F}_t = \delta t(\mathcal{A}_1 - \mathcal{A}_0), \tag{B.4}$$

$$\ell_t(\eta^{(p)}\omega^{(q)}) = (\ell_t\eta^{(p)})\omega^{(q)} + (-1)^p\eta^{(p)}(\ell_t\omega^{(q)}) \tag{B.5}$$

we have,

$$(d\ell_t + \ell_t d)\mathcal{A}_t = \delta t\frac{\partial \mathcal{A}_t}{\partial t} \tag{B.6}$$

$$(d\ell_t + \ell_t d)\mathcal{F}_t = \delta t\frac{\partial \mathcal{F}_t}{\partial t} \tag{B.7}$$

This shows that for any polynomial $S(\mathcal{A}, \mathcal{F})$, we have

$$(d\ell_t + \ell_t d)S(\mathcal{A}_t, \mathcal{F}_t) = \delta t\frac{\partial}{\partial t}S(\mathcal{A}_t, \mathcal{F}_t) \tag{B.8}$$

this yields,

$$S(\mathcal{A}_1, \mathcal{F}_1) - S(\mathcal{A}_0, \mathcal{F}_0) = (dk_{01} + k_{01}d)S(\mathcal{A}_t, \mathcal{F}_t) \tag{B.9}$$

where

$$k_{01}S(\mathcal{A}_t, \mathcal{F}_t) \equiv \int_0^1 \delta t\ell_t S(\mathcal{A}_t, \mathcal{F}_t) \tag{B.10}$$

### 2. Details of gauged WZW term

We would like to use the Carton homotopy method to derive the additional exact form in the gauged WZW term. As discussed around Eq. (IV.3), the general gauged WZW for symmetry breaking of $G \to H$ has the form,

$$\underline{\Gamma}^{(d+1)}(U, A, A_\mathfrak{h}) \equiv \mathsf{CS}(A^U, A_\mathfrak{h}^U) - \mathsf{CS}(A, A_\mathfrak{h}) = \Gamma^{(d+1)}(U) + d\alpha^{(d)}(U, A, A_\mathfrak{h}). \tag{B.11}$$

For $\mathfrak{h} = \varnothing$, the gauged WZW term is given by the Chern-Simons form,

$$\underline{\Gamma}^{(d+1)}(U, A) \equiv \mathsf{CS}(A^U) - \mathsf{CS}(A) = \Gamma^{(d+1)}(U) + d\alpha^{(d)}(U, A). \tag{B.12}$$

For presentation simplicity, we focus on $d = 2$ and first calculate the case when $\mathfrak{h} = \varnothing$, then $\mathfrak{h} \neq \varnothing$.

    *a.*   *The case when $\mathfrak{h} = \varnothing$*   Consider the path of interpolation, $A_t = tU^{-1}AU + U^{-1}dU = tU^{-1}AU + \theta$, such that $A_1 = A^U, A_0 = \theta$. The difference between Chern-Simons forms is then,

$$\mathsf{CS}^{(3)}(A^U) - \mathsf{CS}^{(3)}(\theta) = d\int_t \ell_t \mathsf{CS}^{(3)}(A_t) + \int_t \ell_t d\mathsf{CS}^{(3)}(A_t). \tag{B.13}$$

The last term on the right-hand side (RHS) gives $\mathsf{CS}^{(3)}(A)$, while the first term in the RHS gives,

$$d\int_t \ell_t (A_t F_t - \frac{1}{3}A_t^3) = -d\int_t (A_t A) = d(-dUU^{-1}A), \tag{B.14}$$

since $\ell_t F_t = U^{-1}AU$, $\ell_t A_t = 0$. Therefore, $\alpha^{(2)} = -dUU^{-1}A$. In short,

$$\mathsf{CS}^{(3)}(A^U) - \mathsf{CS}^{(3)}(\theta) = \mathsf{CS}^{(3)}(A) - d(dUU^{-1}A). \tag{B.15}$$

Hence, the gauged WZW term in $d = 2$ is,

$$\underline{\Gamma}^{(3)}(U, A) = \Gamma_G^{(3)}(U) + d(dUU^{-1}A), \tag{B.16}$$

where $\Gamma_G^{(3)}(U)$ is given in Eq. (IV.19). This indeed shows that the gauge field only supports on $d$-dimensional manifold.

    *b.*   *The case when $\mathfrak{h} \neq \varnothing$*   Consider the path of interpolation, $A_t = tU^{-1}AU + \theta, A_{\mathfrak{h},t} = tU^{-1}A_\mathfrak{h}U + V$, the difference of the relative Chern-Simons form is,

$$\mathsf{CS}^{(3)}(A^U, A_\mathfrak{h}^U) - \mathsf{CS}^{(3)}(\theta, V) = d\int_t \ell_t \mathsf{CS}^{(3)}(A_t, A_{\mathfrak{h},t}) + \int_t \ell_t d\mathsf{CS}^{(3)}(A_t, A_{\mathfrak{h},t}). \tag{B.17}$$

Similarly, the last term in the RHS gives $\mathsf{CS}^{(3)}(A, A_\mathfrak{h})$, the first term in RHS is,

$$d\int_t \ell_t \mathrm{tr}\left( (A_t - A_{\mathfrak{h},t})F_t + (A_t - A_{\mathfrak{h},t})F_{\mathfrak{h},t} + ... \right) \tag{B.18}$$

$$= d\int_t \mathrm{tr}(tU^{-1}A_\mathfrak{f}U + \phi)U^{-1}(A + A_\mathfrak{h})U \tag{B.19}$$

$$= d\mathrm{tr}(U\phi U^{-1}(A + A_\mathfrak{h})) \tag{B.20}$$

where $\ell_t F_{t,i} = A_i$, $\ell_t A_i = 0$, and ... is the polynomial $A_t^2 - A_t A_{\mathfrak{h},t}^2 - \frac{1}{3}(A_t^3 - A_{\mathfrak{h},t}^3)$ which vanishes under $\ell_t$. Then the gauged WZW term is given by,

$$\underline{\Gamma}^{(3)}(U, A, A_\mathfrak{h}) = \mathsf{CS}(A^U, A_\mathfrak{h}^U) - \mathsf{CS}(A, A_\mathfrak{h}) = \Gamma^{(3)}(U) + d\mathrm{tr}(U\phi U^{-1}(A + A_\mathfrak{h})) \tag{B.21}$$

where $\Gamma^{(3)}(U)$ is given in Eq. (IV.17).

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
