# Peer review of "Nonlinear sigma model description of deconfined quantum criticality in arbitrary dimensions"

_SciPost Physics_

## Round 2 · Referee Report · Anonymous (Referee 1) · 2022-12-28

Strengths

1 - A nice discussion of various detailed aspects of sigma model approach to deconfined criticality
2 - Extended previous sigma model studies of DQCP (mainly on spherical target spaces) to more general target space G/K, with an application in (3+1)d grand unified theories
3 - Detailed mathematical derivations, especially on the t'Hooft anomalies and the explicit forms of the cohomology generators

Weaknesses

1 - A lot of the physics discussed here seem to be known before, and the author can perhaps make it clearer exactly what is new here. For example, in the discussion on (3+1)d GUT examples, what do we learn from the sigma model approach (beyond what is already there in the literature)?
2 - The discussion on t'Hooft anomaly is a bit confusing. It seems that that relative CS term gives the anomaly only if the spacetime dimension is even? For example, the usual (2+1)d DQCP anomaly (Eq. V. 5) is not related to the CS term? The author should clarify this point.

Report

Overall the manuscript contains some nice discussions (see "Strengths") and could be a useful reference on the sigma model approach to exotic quantum criticality. If the points raised in "Weakness" can be addressed reasonably, the manuscript should be published.

Requested changes

See "Weakness".

  • validity: good
  • significance: ok
  • originality: ok
  • clarity: good
  • formatting: excellent
  • grammar: good

Author:  Da-Chuan Lu  on 2023-04-27  [id 3619]

(in reply to Report 1 on 2022-12-28)
Category:
answer to question

1- In the current manuscript, we construct the coset G/K to describe the given DQCP. In general, the Zn-valued topological defects will become Z valued after the embedding, therefore, it can be explicitly written in the action. For the GUT case, one of the topological defects is Z2 valued, therefore, it is hard to write a differential form for such a defect. One may use Cech cohomology, but it seems hard to do analytical calculations. We embed the topological defect into a large coset G/K, such that the topological defect becomes Z valued and has an associated charge operator written in explicit differential form. We then propose the exotic DQCP among GUTs can be understood by a simple O(6) nonlinear sigma model. The benefits are (1) the duality in the original gauge theory description becomes a global symmetry in the O(6) the nonlinear sigma model, (2) the WZW term assigns phase to the linking between the topological defects, (3) one can do renormalization group analysis using this description. 2- We thank the referee for pointing out this, we are lacking of description in the previous manuscript. Indeed, the relative Chern-Simons term only characterizes the anomaly of even spacetime dimensional theory. For odd spacetime dimensional theory, the anomaly is characterized by the mixed $\theta$ term. The gauged WZW term will be an exact form then. We add necessary clarification and relevant references in the revised manuscript.

---

## Round 2 · Referee Report · Anonymous (Referee 2) · 2023-3-27

Strengths

  1. This paper discusses a detailed procedure to handle the torsion part of the cohomology group of the topological terms in the nonlinear sigma models (NLSMs) and write it in terms of local differential forms by using the embedding method.

  2. Contain some detailed discussion and examples in the context of deconfined quantum critical points (DQCPs).

Weaknesses

  1. It's known that some topological terms can't be written in terms of local differential forms even by using the embedding method and can only be written in terms of some cobordism invariants. The general structure is discussed here: https://arxiv.org/abs/2011.10102. The author should try to comment on this.

Report

This paper provides a nice discussion of the WZW terms in the NLSMs and how to write the topological terms in terms of local differentials by using an embedding method. The discussion is mainly in the context of DQCP which could serve as a nice reference for studying of quantum criticality. Therefore, I recommend the publication of this paper.
  • validity: top
  • significance: ok
  • originality: ok
  • clarity: ok
  • formatting: excellent
  • grammar: good

Author:  Da-Chuan Lu  on 2023-04-27  [id 3618]

(in reply to Report 2 on 2023-03-27)
Category:
answer to question

We thank the referee for pointing out this reference. This reference in general describes the embedding procedure to recover global anomaly cancellation conditions from local anomaly cancellation based on advanced bordism calculation. There are two cases where the embedding procedure could go invalid. (1) the larger group does not contain the relevant irreducible representation that causes the anomaly. As discussed in 5.3 of this reference, the symplectic Majorana multiplet, which is responsible for the 5d SU(2) anomaly, cannot be embedded in U(2). However, if one finds a suitable larger group, then this embedding procedure would still work. (2) the w2w3 anomaly or the new SU(2) anomaly characterizes the global anomaly and is written in bordism invariant. This global anomaly is still present when embedding SU(2) to U(2), also with additional local anomalies. However, the conditions for local anomaly cancellation in the Spin-U(2) theory preclude a non-vanishing new SU(2) anomaly.
In our current manuscript, we use embedding to resolve the homotopy group with discrete generators, such that the homotopy group of the larger space contains integer-valued defects, which can be written in differential forms. This embedding is not directly related to the issues in the aforementioned reference, but it is still worth mentioning. For (1), since G/K contains all the information of topological defects in both symmetry-breaking phases, it is a suitable larger space to consider and reproduce the correct topological charges. For (2), we didn't consider the high codimensional topological defects, so it is unrelated to the 5th-degree bordism group. But for the anomaly matching using the Wess-Zumino-Witten term, this Z2 torsion part is relevant to the sign of the WZW term, and could be determined from the UV theory (please see arXiv:2009.00033, arXiv:2009.04692).

---

## Editorial Decision

resubmitted